# Epigenetic profiling for the molecular classification of metastatic brain tumors

Javier I. J. Orozco [1], Theo A. Knijnenburg[2], Ayla O. Manughian-Peter[1], Matthew P. Salomon[1], Garni Barkhoudarian [3], John R. Jalas[4], James S. Wilmott[5], Parvinder Hothi[6], Xiaowen Wang[1], Yuki Takasumi[4], Michael E. Buckland[7], John F. Thompson[5,8], Georgina V. Long [5,9], Charles S. Cobbs[6], Ilya Shmulevich[2], Daniel F. Kelly[3], Richard A. Scolyer[5,8,10], Dave S.B. Hoon [1,11] & Diego M. Marzese [1]

Optimal treatment of brain metastases is often hindered by limitations in diagnostic capabilities. To meet this challenge, here we profile DNA methylomes of the three most frequent types of brain metastases: melanoma, breast, and lung cancers ($n = 96$). Using supervised machine learning and integration of DNA methylomes from normal, primary, and metastatic tumor specimens ($n = 1860$), we unravel epigenetic signatures specific to each type of metastatic brain tumor and constructed a three-step DNA methylation-based classifier (BrainMETH) that categorizes brain metastases according to the tissue of origin and therapeutically relevant subtypes. BrainMETH predictions are supported by routine histopathologic evaluation. We further characterize and validate the most predictive genomic regions in a large cohort of brain tumors ($n = 165$) using quantitative-methylation-specific PCR. Our study highlights the importance of brain tumor-defining epigenetic alterations, which can be utilized to further develop DNA methylation profiling as a critical tool in the histomolecular stratification of patients with brain metastases.

[1] Department of Translational Molecular Medicine, John Wayne Cancer Institute at Providence Saint John's Health Center, Santa Monica, CA 90404, USA. [2] Institute for Systems Biology, Seattle, WA 98109, USA. [3] Pacific Neuroscience Institute, John Wayne Cancer Institute at Providence Saint John's Health Center, Santa Monica, CA 90404, USA. [4] Department of Pathology, Providence Saint John's Health Center, Santa Monica, CA 90404, USA. [5] Melanoma Institute Australia, The University of Sydney, Sydney, NSW 2065, Australia. [6] Ben & Catherine Ivy Center for Advanced Brain Tumor Treatment, Swedish Neuroscience Institute, Seattle, WA 98122, USA. [7] Department of Neuropathology, Royal Prince Alfred Hospital, the Brain and Mind Centre, The University of Sydney, Camperdown, NSW 2050, Australia. [8] Sydney Medical School, The University of Sydney, Camperdown, NSW 2006, Australia. [9] Royal North Shore Hospital, Sydney, NSW 2065, Australia. [10] Royal Prince Alfred Hospital, Sydney, NSW 2050, Australia. [11] Sequencing Center, John Wayne Cancer Institute at Providence Saint John's Health Center, Santa Monica, CA 90404, USA. Correspondence and requests for materials should be addressed to D.M.M. (email: MarzeseD@jwci.org)

Brain metastases (BM) are the most common intracranial neoplasm in adults and are the next frontier for the management of metastatic cancer patients. Large population-based studies have shown that 8–10% of cancer patients develop BM, with this proportion increasing up to 26% when autopsy studies were included[1–5]. Lung cancer, breast cancer, and cutaneous melanoma account for the vast majority (75–90%) of secondary neoplasms in the brain[1–4]. Treatment options for BM include surgery, whole-brain radiotherapy, stereotactic radiosurgery, and systemic drug therapy, such as chemotherapy, targeted therapies, and immunotherapy[6]. While systemic chemotherapy has limited efficacy, targeted therapies have recently shown promise for oncologic management[6]. These tailored therapies have significantly affected treatment decision-making for patients with breast cancer BM (BCBM). For example, patients with human epidermal growth factor receptor 2 (HER2)-positive BCBM can be treated with anti-HER2 agents[7] and patients with estrogen receptor (ER)-positive BCBM with endocrine agents, cyclin-dependent kinases 4 and 6 (CDK4/6) inhibitors, and the mechanistic target of rapamycin kinase (mTOR) inhibitors[8]. As such, accurate diagnosis is essential to effectively treat patients with BM.

Diagnosis of BM is currently based on neuro-imaging and confirmed by pathology examination. When appropriate, the diagnostic algorithm begins by distinguishing BM from primary brain tumors using histologic features guided by the clinical and radiologic information[9]. Then, to identify the tissue of origin, morphological evaluation is supplemented by several immunohistochemistry (IHC) markers including thyroid transcription factor (TTF-1), chromogranin and synaptophysin for lung cancer BM (LCBM); GATA3 binding protein (GATA3), mammaglobin, gross cystic disease fluid protein 15 (GCDFP-15) and ER for BCBM; and human melanoma black 45 (HMB45), melanoma antigen recognized by T-cells 1 (Melan A/MART-1), SRY-Box 10 (SOX-10), and the S100 calcium binding proteins (S-100) for melanoma BM (MBM)[9,10]. However, a major limitation in achieving an accurate pathological diagnosis is the often poor differentiation and/or limited availability of metastatic brain tumor tissues to evaluate the complete panel of IHC markers[9].

The advent of molecular classifiers based on the synergy between comprehensive tumor profiling and statistical modeling has dramatically improved the diagnosis, prognosis and, importantly, the therapeutic approaches for cancer patients[11]. To date, however, molecular classifiers to assist pathological diagnosis and improve stratification of patients with metastatic brain tumors have been ill-defined. DNA methylation (DNAm) profiling was recently shown to be a powerful analytical tool to identify the origin of cancers from unknown primary sites and to better stratify patients with primary central nervous system (CNS) tumors[12,13]. We additionally have shown that DNAm profiling can be efficiently performed using small samples of BM tissues[14–17].

Here, we hypothesize that the construction and validation of DNAm classifiers for BM could address current anatomic pathology diagnostic issues. The objective of our study is to identify genomic regions whose DNAm status allow for (i) discrimination between primary and metastatic brain tumors, (ii) accurate identification of the tissue of origin for metastatic brain tumors, and (iii) assistance in the classification of therapeutically relevant subtypes for patients with BCBM. For this study, we included 165 patients with surgically resectable primary or metastatic brain tumors. Using Infinium HumanMethylation 450K (HM450K) microarray technology we generate high-quality genome-wide DNA methylomes for 96 microdissected BM specimens including BCBM ($n = 30$), LCBM ($n = 18$), MBM ($n = 44$), and BM with uncertain histogenesis ($n = 4$). We further integrate our data with additional publicly available DNA methylomes ($n = 1860$) to construct and evaluate a robust brain metastasis DNAm classifier (BrainMETH). BrainMETH involves a three-step classification process that assists in diagnosing brain neoplasms: first, by discriminating between primary and metastatic brain tumors (Class A), second, by identifying the tissue of origin of the BM (Class B), and finally, by discriminating BCBM subtypes (Class C). We additionally use BrainMETH to identify the tumor site of origin of BM with an 'uncertain' diagnosis and to predict the status of IHC markers of BCBM that were not assessed in the initial diagnosis. Importantly, highlighting its potential clinical utility, we show that targeted quantitative-methylation-specific PCR (qMSP) can be applied to efficiently evaluate the representative genomic regions included in BrainMETH in DNA extracted from microdissected formalin-fixed paraffin-embedded (FFPE) archived tissue sections.

## Results

**Brain metastasis DNA methylation data processing**. To identify intrinsic differences in the epigenetic profiles of metastatic brain tumors, we first generated DNAm signatures for 96 microdissected BM tissues (a list of BM specimens with clinical and demographic information can be found in Supplementary Note 1 and Supplementary Data 1). This comprehensive profiling was performed using the HM450K microarray, which, in addition to currently being the most commonly used DNAm profiling platform, has also been employed by The Cancer Genome Atlas (TCGA) project to profile large cohorts of solid tumors[18,19]. Based on the most recent characterization of HM450K probes[20], 102,941 genomic regions were excluded from downstream analyses. The exclusion criteria included probes that recognize common single nucleotide polymorphisms (SNPs), repetitive genomic elements, high GC density (> 25%) on the 50-bp length probe sequence, a non-unique mapping to the genome, or low mapping quality (Fig. 1a). Additionally, to decrease potential biases associated with the gender of BM patients, we excluded 105,422 probes recognizing regions located in sex chromosomes or with proven cross-reactivity with sex chromosomes[21,22]. 2983 probes with a detection $P$-value greater than 0.01 ('NA') in any specimen were also excluded from this study. In addition, to obtain a set of genomic regions comparable to the newer generation of DNAm arrays, we excluded 13,882 probes that have been removed from the design of the HumanMethylation EPIC BeadChip array[23]. Finally, to decrease the influence of non-tumor cells, we excluded 48,797 genomic regions with no significant DNAm differences (Wilcoxon test; $P$-value > 0.05) between BMs and normal brain tissues ($n = 100$; GSE43414). Thus, this pipeline identified a dataset containing 211,552 informative HM450K probes to explore similarities and differences among brain tumors (Fig. 1a).

**DNAm profiles of primary and metastatic brain tumors**. Single intracranial metastases and primary brain tumors, mainly high-grade gliomas or glioblastomas (GBM), often exhibit overlapping clinical and radiological features[24]. To evaluate the potential utility of DNAm profiling for the classification of brain tumors, we identified a cohort of patients with GBM whose tumors have been profiled using the HM450K platform by TCGA-GBM project[25]. In order to reduce the impact of the lack of tissue microdissection on TCGA samples, we excluded GBM specimens with < 70% tumor purity evaluated by the consensus purity estimation (CPE) method[26]. As expected, correlation and principal component analyses (PCA) using randomly-selected HM450K probe sets (mean number of probes per set = 22,543 ± 150.7; Supplementary Data 2) revealed significant DNAm

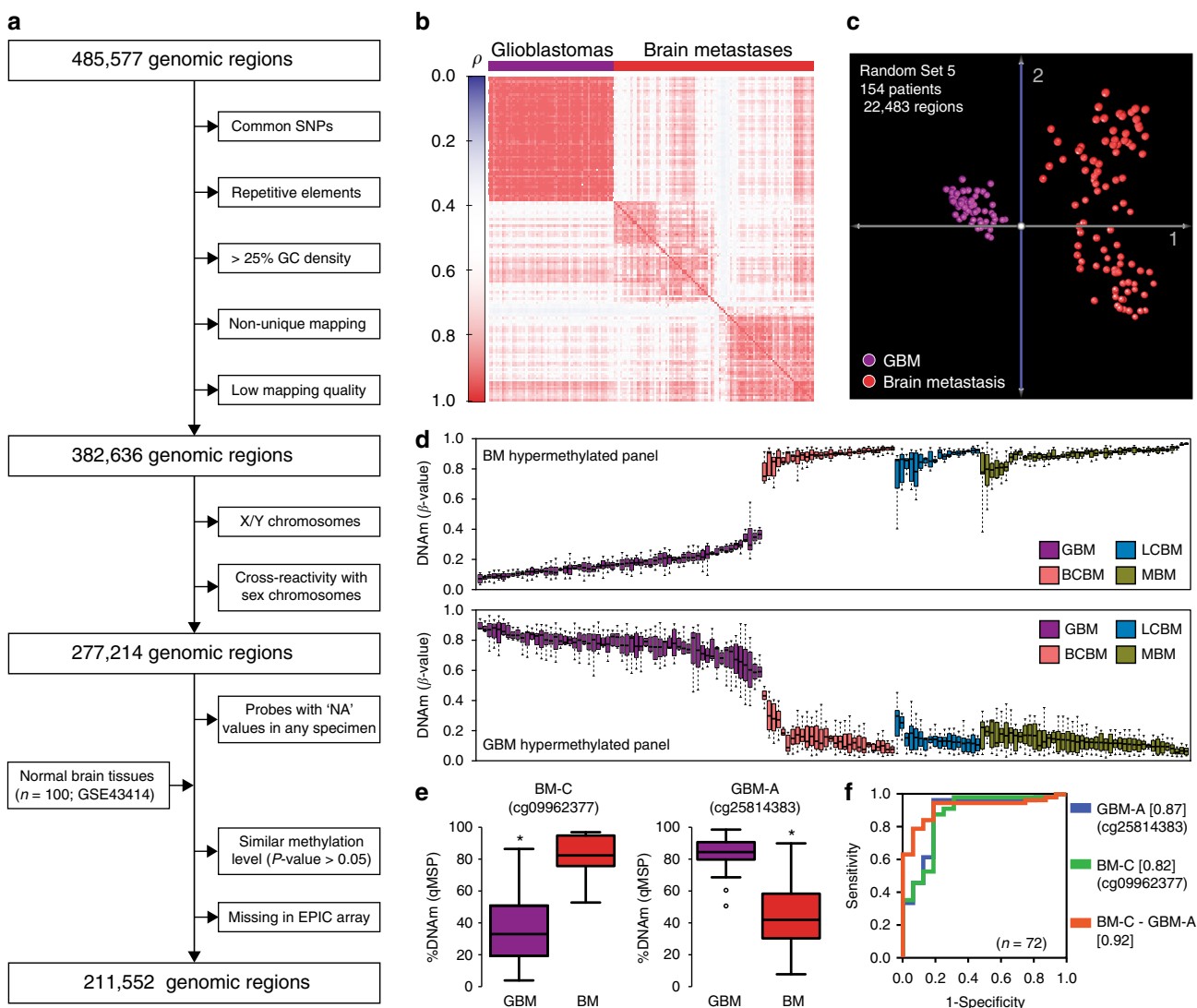

**Fig. 1** Genome-wide DNA methylation profiling of brain tumors. **a** Diagram describing the normalization algorithm of Infinium HM450K probes. **b** Matrix depicting the Spearman's $\rho$ correlation coefficients among primary and metastatic brain tumors. **c** Principal component analysis (PCA) of GBM ($n = 60$) and brain metastases ($n = 94$) using DNA methylation level of 22,483 randomly-selected genomic regions. **d** DNA methylation level of the 12 most differentially methylated regions between GBM and BM specimens. The upper panel shows the $\beta$-value of six genomic regions differentially hypermethylated in BM (cg07076109, cg19111287, cg09962377, cg10982851, cg15002250, and cg23108580) and the lower panel shows the $\beta$-value of six genomic regions differentially hypermethylated in GBM (cg25814383, cg26306329, cg06663644, cg20604286, cg04314308, and cg26306994) for each specimen in the study ($n = 154$). The top and bottom of each box represent the first and third quartile, respectively; the internal line represents the median. **e** Validation of differentially methylated regions by qMSP in an independent cohort of brain tumor specimens ($n = 72$). The left boxplot shows the methylation level of a genomic region hypermethylated in BM specimens (BM-C; cg09962377) and the right boxplot the level of a genomic region hypermethylated in GBM specimens (GBM-A; cg25814383; chr19:19,336,240). The top and bottom of each box represent the first and third quartile, respectively; the internal line represents the median. *Wilcoxon test, $P$-value < 0.02. **f** ROC showing the prediction potential of brain tumor type using qMSP scores for each independent genomic region and score for the combination (DNAm level of BM-C minus DNAm level of GBM-A; AUC = 0.92, 95% CI = 0.85-0.99); see Supplementary Data 3 for details about validated genomic regions. The AUC values are indicated between square brackets

differences between primary and metastatic brain tumors (Fig. 1b, c). The first three components of the PCA explained a mean of 84.14 ± 0.6% of the cumulative variance (Supplementary Data 2). GBM specimens showed a more confined distribution in the three-dimensional component space of the PCA than the BM specimens (Fig. 1c), which was also observed when comparing GBM specimens with each of the BM types (BCBM, LCBM, and MBM; Supplementary Fig. 1a). Interestingly, by analyzing a recently published DNAm dataset (GSE90496; $n = 2085$)[13], we found that DNAm profiles of BMs substantially differ from a wide range of primary CNS tumors. Supporting this finding,

distinct clustering of BM specimens can be observed in the t-distributed stochastic neighbor embedding (t-SNE) and TumorMap visualizations (Supplementary Fig. 1b, c). Based on the intrinsic differences in DNAm distributions of primary versus metastatic brain tumors, we explored differentially methylated regions with the potential to discriminate between these tumor types. Employing a strict statistical cut-off (absolute DNAm difference > 30%; false discovery rate (FDR)-corrected $q$-value < 0.01), we identified 14,494 regions differentially methylated between GBM and BM specimens (Supplementary Fig. 1d). The Genomic Regions Enrichment Annotations Tool (GREAT)[27]

indicated that regions hypomethylated in GBM ($n = 8905$) were predominantly associated with genes involved in neuronal differentiation and proliferation (Hypergeometric test; FDR-corrected $q$-value $= 1.96e{-}39$) and regions hypomethylated in BM ($n = 5589$) were associated with genes involved in neuronal formation, development, and differentiation (Hypergeometric test; FDR-corrected $q$-value $= 1.14e{-}10$; Supplementary Fig. 1e). Twelve differentially methylated regions were selected based on a low overall variance and large absolute differences in DNAm levels between GBM and BM types. Six were consistently hypermethylated in BM specimens (called herein BM-A to BM-F), and six were hypermethylated in GBM specimens (called herein GBM-A to GBM-F; Fig. 1d and Supplementary Fig. 2a). Using the HM450K data, we found that by surveying at least two of these genomic regions hypermethylated in BM we could distinguish primary from metastatic brain tumors with significant accuracy (Area under the curve (AUC) > 0.90; Supplementary Fig. 2b). Importantly, discrimination power was improved when combining regions with opposing DNAm patterns in GBM and in BM specimens. The evaluation of one BM hypermethylated region with one GBM hypermethylated region showed higher prediction capability (AUC = 0.99, 95% CI = 0.99–1.00) than evaluation of single genomic regions (Supplementary Fig. 2c). We, therefore, assessed the classification potential of these regions in independent cohorts ($n = 227$) of patients with GBM (GSE85539)[28], patients with low-grade gliomas (LGGs) from the EORTC-26951 phase III clinical trial (GSE48461)[29], and patients with BM (GSE44661) included in our previous studies[15]. We found that evaluation of single genomic regions provides substantial discrimination potential between BM and GBM ($n = 168$; AUC range = 0.97–0.99; Supplementary Fig. 3a) and moderate discrimination efficiency between BM and LGG ($n = 75$; AUC range = 0.67–0.89; Supplementary Fig. 3b). To further evaluate the pairing of BM with GBM hypermethylated regions, we analyzed the DNAm level of these 12 genomic regions in an independent cohort of microdissected brain tumor clinical specimens ($n = 72$) using qMSP. This analysis confirmed the significant DNAm differences of these regions between GBM and BM specimens (Wilcoxon test; $P$-value < 0.02; Fig. 1e). Thus, based on qMSP evaluation, we identified regions with poor, moderate, and good performance (genomic coordinates, nearby genes, primer sequences, and qMSP performance can be found in Supplementary Data 3). While using additional qMSP primer sets did not improve the performance of six poor performing regions, we were able to determine the DNAm status by employing locus-specific bisulfite sequencing (Supplementary Fig. 3c; sequencing primer sequences listed in Supplementary Data 3). However, because of the quantitative nature of qMSP, we employed this method to analyze regions with good and moderate qMSP performance and thus generate a scale variable for assessing accuracy to discriminate between types of brain tumors. We found that DNAm level of one region could accurately distinguish BM from GBM specimens (i.e. for GBM-A: AUC = 0.87, 95% CI = 0.76–0.98). Importantly, the combination of two regions with good qMSP performance (BM score = DNAm level of BM-C minus DNAm level of GBM-A; Supplementary Fig. 3d) showed a higher prediction potential than single regions (AUC = 0.92, 95% CI = 0.85–0.99; Fig. 1f). Based on these results, regions whose DNAm level (as assessed by qMSP) exhibited good predictive accuracy, were selected as the first step of the DNAm-based brain tumor classifier (BrainMETH class A).

**DNAm profiles of brain metastases from different origins**. We observed that BM DNAm profiles present significantly lower overall correlation (mean Spearman's $\rho = 0.77 \pm 0.02$) compared

to GBM (mean Spearman's $\rho = 0.90 \pm 0.06$; $P$-value $= 2.2e{-}16$; Fig. 2a and Supplementary Fig. 4a). Biologically, this epigenetic variability may reflect the diversity of the tissues of origin and the influence of differing pre-BM therapeutic approaches. Based on these observations, we further explored differences among DNAm signatures of intracranial metastases. Interestingly, unsupervised hierarchical cluster (HCL) analysis of the top 5000 most variable regions precisely separated the BM specimens into two main clusters (Bootstrap value = 100%; Fig. 2b). The influence of the epigenetic landscape of the tissue of origin was reflected in the organization of the hierarchical tree. Cluster A included BMs from patients with epithelial tumors (breast and lung carcinomas) and cluster B contained BMs from patients with neuroectodermal tumors (cutaneous melanoma; Fig. 2b and Supplementary Fig. 4b). Similar results were observed even when the number of genomic regions was increased from 5000 to 100,000 (Supplementary Fig. 4c–f). By grouping all the BM specimens according to the pathologically confirmed primary tumor of origin ($n = 90$), we found 31,818 regions differentially methylated among the three BM types (one-way ANOVA; Bonferroni adjusted $P$-value < 0.05; Supplementary Data 4). Pathway enrichment analyses of BM type-specific hypomethylated regions revealed multiple relevant genes and pathways specific to each BM type, including upregulation of serpine molecules, which has been linked to the ability of BM cells to prevent attack by reactive astrocytes[30], and enhanced syndecan-mediated signaling events, which have been implicated in promoting BM cell migration through the blood-brain barrier[31] (Supplementary Data 5). PCA of the differentially methylated genomic regions ($n = 31,818$) showed a clear separation of BM specimens according to the tumor of origin (Fig. 2c). The distribution of each BM type suggested that the various treatment modalities employed preceding our analysis did not significantly affect the DNAm levels of these genomic regions. For this comparative analysis, we considered any prior systemic therapy (PST), prior chemotherapy (PCT), prior targeted therapy (PTT), or prior radiotherapy (PRT; Supplementary Fig. 5). We found a negligible number of genomic regions to be significantly associated with prior treatment modalities (Bonferroni adjusted $P$-value < 0.05; mean = 14.4, range: 0–91; < 0.02% of the assayed regions). Then, using the 31,818 differentially methylated genomic regions, we evaluated BMs from four female patients treated for LCBM, but with ambiguous IHC profiling, or with a history of both primary lung cancer and primary breast cancer. In accordance with the pathological presumptive diagnosis, these four BM specimens with "uncertain" diagnosis showed an overlap with pathologically confirmed LCBM specimens (Fig. 2d). Moreover, we found that the spatial separations based on tumor DNAm signatures between BCBM, LCBM, and MBM specimens were independent of the patient's gender, as demonstrated by the analysis of a sub-cohort of BMs from female patients ($n = 58$; Fig. 2e). Together, these results reflect the efficiency and potential utility of DNAm profiling in accurately identifying the tissue of origin of intracranial metastases.

**DNAm classifiers identify the origin of brain metastases**. Based on the observed differences in methylation patterns among LCBM, BCBM, and MBM specimens, we constructed and evaluated DNAm classifiers to efficiently identify the BM tissue of origin using a random forest (RF)-based supervised learning approach[32]. We initially used the top 10,000 most variable differentially methylated regions among BCBM, LCBM, and MBM specimens. Overall, the resulting classifiers demonstrated an excellent classification potential (Fig. 3a) with an average sensitivity and specificity over 90% for all three BM types (MBM, BCBM, and LCBM;

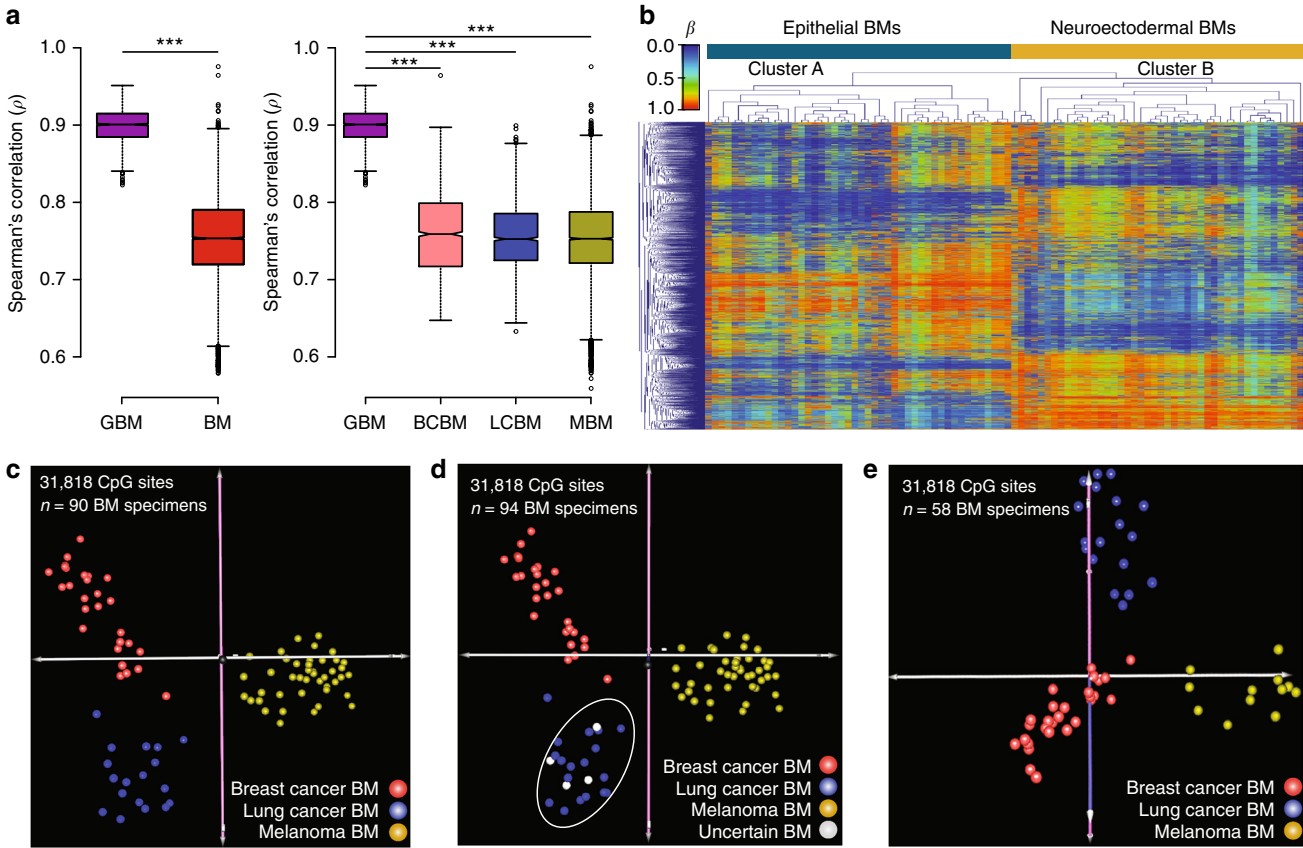

**Fig. 2** DNA methylation differences among brain metastases. **a** Boxplot representing overall Spearman's $\rho$ correlation coefficients among brain tumors. The left plot includes all the GBM specimens ($n = 60$, mean $\rho = 0.90 \pm 0.06$) and all the BM specimens ($n = 94$, mean $\rho = 0.77 \pm 0.02$). The right plot describes the overall Spearman's $\rho$ correlation coefficient among BM specimens with anatomical pathology confirmed tumor of origin (BCBM $n = 28$, LCBM $n = 18$, and MBM $n = 44$). The top and bottom of each box represent the first and third quartile, respectively; the internal line represents the median. ***Spearman's correlation test; $P$-value < 0.001. **b** Unsupervised hierarchical clustering using Euclidean distance of the top 5000 most variable genomic regions. **c** PCA using 31,818 CpG sites with significant (ANOVA, Bonferroni adjusted $P$-value < 0.05; Supplementary Data 4) differential DNA methylation level among BM with anatomical pathology confirmed tissue of origin ($n = 90$). **d** PCA using the differentially methylated region including four BM specimens with uncertain primary tumor of origin ($n = 94$). **e** PCA including BM specimens from female patients ($n = 58$)

Fig. 3b). We found that by surveying as few as 20 regions, the classifiers exhibited a median cross-validation (CV) performance above 90%, with a deterioration of this value observed only when employing less than 10 regions (Fig. 3a). Thus, we identified the regions with the highest importance for the prediction of the tumor of origin (Gini impurity score (GIS); Fig. 3c). Additionally, to better understand the basis of the DNAm signatures that stratify BM specimens by the tumor of origin, DNA methylomes from breast, lung, and melanoma primary tumors generated by TCGA projects were used to test the prediction performance of these same regions when applied to primary tumors. Overall, we found that patterns of differential methylation of these regions for BMs and primary tumors were in agreement (Fig. 3d). Specifically, the top 100 most informative BM regions showed good performance for the classification of primary tumors according to the tumor type. The first three components of the PCA explained up to 75.5% of the cumulative variance (Supplementary Fig. 6a). Bootstrap resampling of the HCL showed 100% support for the separation between the cluster containing the primary melanomas and the cluster containing the primary breast and lung carcinomas, and 78% support for the separation between the cluster containing most of the primary breast tumors and the cluster containing most of the primary lung tumor specimens (Supplementary Fig. 6b). Moreover, an independent RF classification model applied to the primary tumors using TCGA DNAm data

revealed a highly significant overlap in the top 100 most predictive genomic regions between the BM and the primary tumor classifiers (Hypergeometric test; $P$-value < 2.8e−23). These findings suggest that BM type-specific DNAm signatures are comparable to genomic region differences between their corresponding primary tumors. To further examine the ability of these regions to classify BM tumor of origin, we then refined the number of regions by selecting nine which exhibited a low overall variance within each tumor type, and a large difference in the mean DNAm level among the three BM types (Supplementary Fig. 6c–e). Individually, DNAm levels of these regions demonstrated good performance in identifying the BM tumor of origin ($n = 94$; Supplementary Fig. 7a). We therefore designed qMSP assays for each region and evaluated DNAm levels in metastatic brain tumor clinical specimens ($n = 59$). Based on these results, each assay was categorized into good, moderate, and poor qMSP performance (genomic coordinates, nearby genes, primer sequences, and qMSP performance can be found in Supplementary Data 6). DNAm status of regions exhibiting poor performance was established using locus-specific bisulfite sequencing (Supplementary Fig. 7b; sequencing primer sequences listed in Supplementary Data 6). We then selected three regions, one per BM type, with a significant correlation between qMSP and HM450K assays (Spearman's $\rho$; $P$-value < 0.001; Supplementary Fig. 7c) and significant differential methylation among the BM types (Wilcoxon test; $P$-value < 0.001;

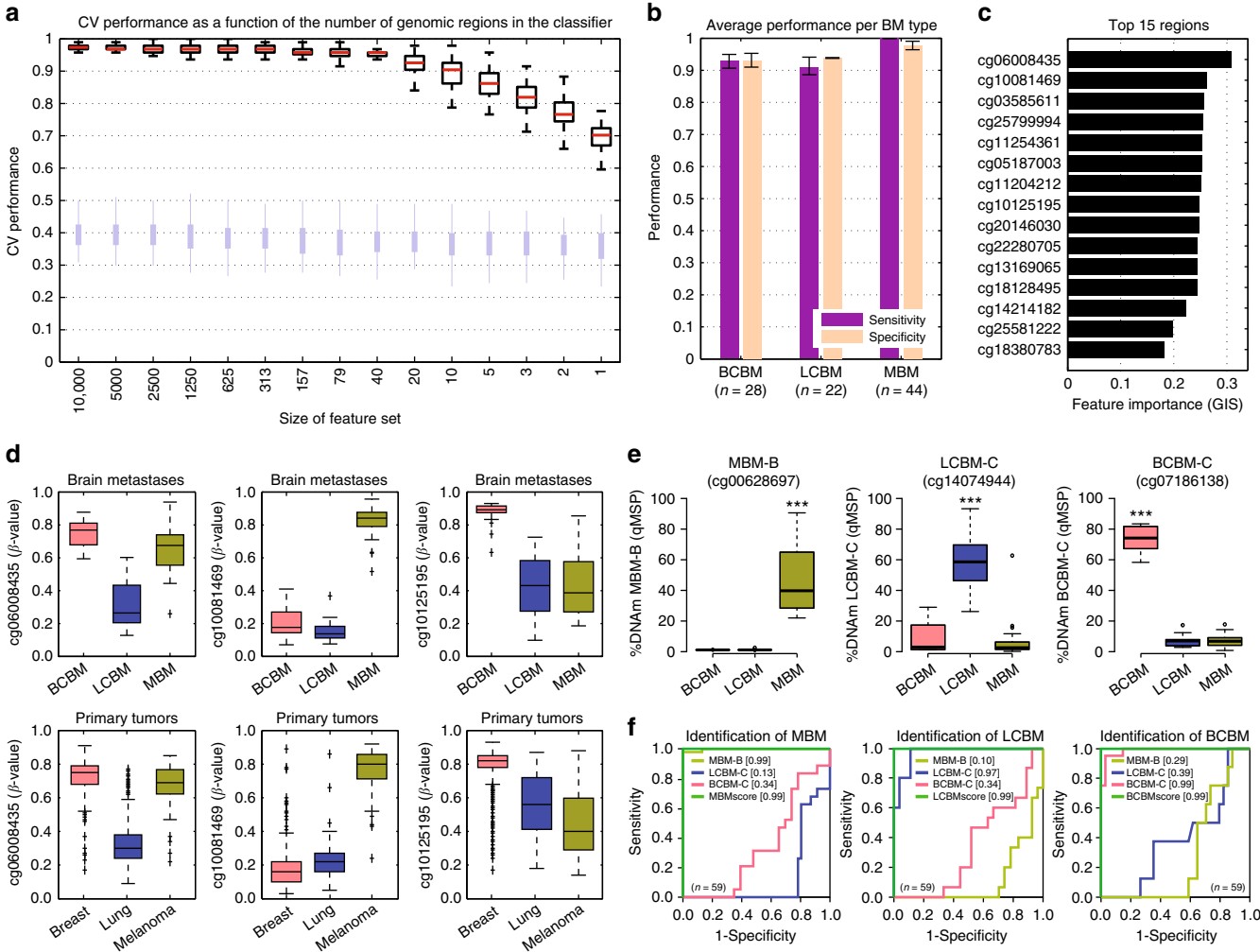

**Fig. 3** DNA methylation classifiers to predict the tissue of origin of brain metastases. **a** CV performance across 100 repeats for the RF classifiers to predict tumor of origin, in order of decreasing number of features used in model construction, from left to right (x-axis). Red bars on the boxplots indicate medians and light blue bars depict the performance based on permuted class labels and represent the random background distribution. **b** Bar plots depicting the prediction performance as measured by sensitivity and specificity for each of the BM types. The bars show the average performance and interquartile range (error bars) across all models with 40 features or more across all repeats. **c** Bar plots depicting the RF feature importance (mean decreases in Gini impurity score; GIS) of the 15 most predictive genomic regions averaged across all models with 40 features or more and across all repeats. **d** For three genomic regions in the top 15: boxplots of DNAm β-values across our cohort stratified by tumor of origin (BCBM n = 28, LCBM n = 22, and MBM n = 44) in the upper panels and TCGA cohorts of primary breast tumors (n = 401), primary lung tumors (n = 307), and primary melanomas (n = 83) in the lower panels. Differences in the DNAm levels among the groups were statistically significant for all the cases (Kruskal–Wallis test; P-value < 0.0001). **e** DNAm levels assessed by qMSP for three regions differentially methylated among the three BM types (n = 59). The top and bottom of each box represent the first and third quartile, respectively; the internal line represents the median. ***Wilcoxon test; P-value < 0.001. **f** ROC curves showing the prediction potential for the tumor of origin (n = 59) for each of the differentially methylated regions and combinations into BM type-specific scores: MBMscore = DNAm level of MBM-B minus DNAm level of LCBM-C minus DNAm level of BCBM-C; LCBMscore = DNAm level of LCBM-C minus DNAm level of BCBM-C minus DNAm level of MBM-B; and BCBMscore = DNAm level of BCBM-C minus DNAm level of LCBM-C minus DNAm level of MBM-B; see Supplementary Data 6 for details about these genomic regions. The AUC values are indicated between square brackets

Fig. 3e). qMSP evaluation of these regions showed excellent prediction performance to identify the tissue of origin (AUC range = 0.97–0.99), which was slightly improved by combining one region per BM type into a tumor origin-specific DNAm score (mean AUC = 0.99; Fig. 3f). As evidence of the potential practical applications of these qMSP assays, we found that the evaluation of three regions (MBM-B, LCBM-C, and BCBM-C) confirmed the origin of four BMs with uncertain diagnosis as LCBM, as forecasted by HM450K data (Supplementary Fig. 7d). Thus, genomic regions with good qMSP classification performance were selected as the second step of the brain tumor DNAm classifier (Brain-METH class B).

**Differential DNAm patterns among BCBM subtypes.** Characterization of BCBM subtype is critical to guiding clinical management. Therefore, we further explored the ability of DNAm profiling to discriminate between molecular subtypes of BCBM specimens. The expression levels of the hormone receptors (HR; ER and progesterone (PgR)) and HER2 assessed by IHC, allowed us to classify 24 of the 28 BCBMs into three clinically relevant classes: (1) HR+/HER2−, (2) HR-positive or HR-negative/HER2-positive (labeled HER2+), and (3) HR−/HER2−. The first evaluation involved the identification of regions differentially methylated among the three BCBM subtypes. Because patient gender was homogeneous, we included 4229 additional good

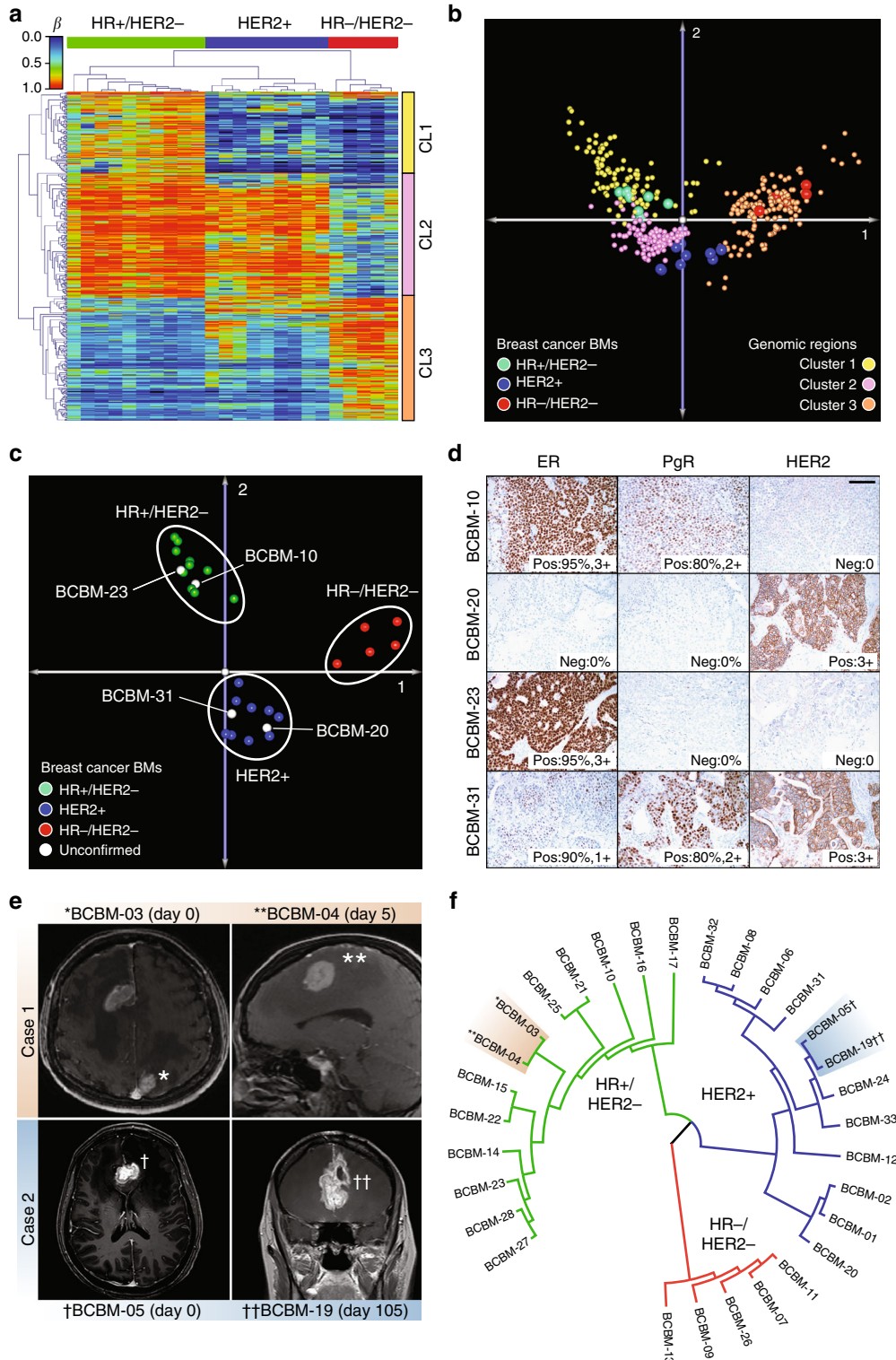

quality probes recognizing genomic regions located on the X-chromosome. HCL using 409 significantly differentially methylated regions (one-way ANOVA; FDR-corrected $q$-value < 0.0005; Supplementary Data 7) generated three sub-clusters containing each of the BCBM subtypes (Bootstrap value = 100%; Fig. 4a). Interestingly, HER2+ and HR+/HER2− BCBM specimens, which are generated from the late and differentiated luminal progenitor mammary cells, respectively, were included in a

common cluster, while HR−/HER2− BCBM specimens, which commonly present gene expression profiles similar to mammary myoepithelial (basal layer) cells, were included in a separate cluster, indicating the relationship between the DNAm landscapes and the cell type of origin[33,34], as well as supporting the concept that basal-like breast cancer represents a unique disease entity[35]. We found that these differentially methylated regions were organized in three specific clusters (Fig. 4a and

**Fig. 4** DNA methylation differences among BCBM subtypes. **a** Hierarchical cluster analysis using Euclidean distance for the DNAm level of 409 regions significantly differentially methylated (one-way ANOVA; FDR-corrected $q$-value < 0.0005; Supplementary Data 7) among the three breast cancer molecular subtypes ($n = 24$ BCBM specimens). This analysis revealed three distinct clusters of genomic regions. CL1 includes regions specifically methylated in HR+/HER2− BCBM, CL2 includes regions methylated in both, HR+/HER2− BCBM and HER2+ BCBM, and CL3 includes regions specifically methylated in HR−/HER2− BCBM. **b** Two-dimensional projection depicting the DCA for differentially methylated regions ($n = 409$) and BCBM specimens ($n = 24$) with known IHC profile. This plot shows the spatial overlapping of genomic regions with relative importance for each BCBM molecular subtype. **c** PCA including 24 BCBM specimens with known molecular subtypes and four BCBM with missing IHC information using 126 genomic regions with classification potential. The unconfirmed specimens were assigned to two different clusters. BCBM-10 and BCBM-23 overlapped with HR+/HER2− BCBM and BCBM-20 and BCBM-31 overlapped with HER2+ BCBM. **d** IHC evaluation in a CLIA-certified Pathology Department for ER, PgR, and HER2 expression (scale bar, 100 μm). The results confirm the DNAm-based prediction for the expression of HR and HER2. **e** Magnetic resonance imaging showing two patients with synchronous (case 1) BCBM lesions (BCBM-03 and BCBM-04) and asynchronous (case 2) BCBM lesions (BCBM-05 and BCBM-19). **f** Phylogenetic tree generated using the Euclidian metric distance for BCBM according to DNAm profile of the 126 genomic regions

Supplementary Data 8). The first cluster (CL1) included regions hypermethylated in HR+/HER2− BCBM and the third cluster (CL3) included regions hypermethylated in HR−/HER2− BCBM, while the second cluster (CL2) included regions hypermethylated in either HR+/HER2− or HER2+ BCBM and hypomethylated in HR−/HER2− BCBM specimens. This organization further highlights the relationship between DNAm profiles and the originating cell type (Supplementary Data 8). Due in part to the small number of differentially methylated regions among the BCBM molecular subtypes, the GREAT analysis did not indicate significant enrichment of genes and pathways for regions specifically methylated in each BCBM subtype (threshold: hypergeometric test; FDR-corrected $q$-value = 0.05). We also explored the associations between the three clusters of genomic regions and the specific BCBM subtypes using a detrended correspondence analysis (DCA; cumulative inertia of the first three axes = 80.1%; Fig. 4b). This analysis showed that not all these regions are equally relevant to define the BCBM subtypes. Therefore, by employing the nearest shrunken centroid algorithm, we identified 126 regions specifically associated with BCBM molecular subtype ($\delta = 2.1$ and $\rho = 0.9$; Supplementary Data 9). To gauge the ability of DNAm profiling to predict BCBM subtypes, we included four additional patients without prior HR and/or HER2 IHC evaluation of their BM specimens, which were treated based on IHC evaluation of their primary breast cancer tumors and/or extra-cranial metastases. Multidimensional reduction analysis using the DNAm level of the 126 regions suggested that two of these BCBM specimens (BCBM-10 and BCBM-23) belong to the HR+/HER2- subtype and the other two (BCBM-20 and BCBM-31) belong to the HER2+ subtype (Fig. 4c). Remarkably, the BCBM subtypes predicted by the DNAm analysis were prospectively confirmed by IHC analyses of ER, PgR, and HER2 by the Department of Pathology at Providence Saint John's Health Center (Fig. 4d). In view of these promising results, we then evaluated the utility of this epigenetic signature to recognize multiple BM lesions. To this end, we performed HM450K profiling of two cases with multiple BCBM. The first patient presented two synchronous metastatic brain tumors, and the second patient presented asynchronous metastatic lesions (BCBM-03 paired with BCBM-04 and BCBM-05 paired with BCBM-19, respectively; Fig. 4e). Interestingly, HCL of the 126 identified regions (Supplementary Data 9) denoted an overlap between paired BM lesions. These results suggest that the BCBM subtypes can be inferred from the DNAm levels using a relatively small set of genomic regions.

**Assembly of DNAm classifiers to identify BCBM subtypes.** Due to the substantially different DNAm signatures observed among BCBM subtypes, we set out to build a classification scheme that could accurately recognize therapeutically oriented BCBM

subtypes. Starting with the top 10,000 most variable regions in the BCBM dataset, we employed an RF-based approach[32] to train and test a classifier that discriminates between the three BCBM subtypes. All the resulting classifiers showed improved classification performance compared to random classification (Fig. 5a), yet substantially decreased performance compared to the DNAm classifiers generated to identify BM tissue of origin (Fig. 3a). This difference in the classification performances can be explained by two facts: (1) the sample size to train the BM tissue of origin classifier is larger than the BCBM subtype classifier ($n = 94$ vs $n = 28$, respectively), enabling more complex and predictive patterns to be learned from the data and (2) there is a significantly larger difference in global DNAm, and thus discriminatory power, between BM originating from different tissues than between different BCBM subtypes. Here, this was clearly evidenced by the larger variance explained by BM types compared to BCBM subtypes (mean F-Ratio for BM types: $10.02 \pm 16.4$ and mean F-Ratio for BCBM subtypes: $2.29 \pm 3.35$; Wilcoxon test; $P$-value < 2.2e−16). In spite of these technical and sample limitations, classifiers using few genomic regions (between 10 and 40) retained a good overall classification performance, similar to classifiers using the entire dataset (Fig. 5a). Specifically, the median CV performance started to deteriorate only with classifiers comprised of fewer than 10 genomic regions. In keeping with our observation of the effect of sample size on classifier performance, the most frequent BCBM subtype (HR+/HER2−), could be predicted with the highest sensitivity and specificity (Fig. 5b). Additionally, regions with high feature importance (GIS; Fig. 5c) were compared with DNA methylomes from primary breast cancer tumors generated by TCGA-BRCA project. The DNAm patterns of the most informative regions were generally in agreement between BCBM and primary breast cancer specimens (Fig. 5d). However, unlike our observation for the BM tissue of origin classifiers, the genomic locations that efficiently classified BCBM subtypes, showed modest to poor efficiency for classification of primary breast cancer specimens into subtypes (Supplementary Fig. 8a). PCA using the top 100, 50, 30, 15, 10, and 5 most informative BCBM regions, explained a mean of $58.50 \pm 3.2\%$ of the cumulative variance for the distribution of primary breast cancer specimens. This result suggests that the selected regions may be exclusively relevant for BCBM classification and should not be employed to discriminate among primary breast cancer molecular subtypes. It also suggests that regions discriminating BCBM subtypes may be determined or influenced by the tissue environment, and potentially, the treatment regimens. To explore the potential practical applications of the identified DNAm signatures, we aimed to efficiently classify BCBM according to molecular subtypes. To this end, here we selected ten regions with low variance within each subtype, and large DNAm differences among the three classes (Supplementary Fig. 8b–d).

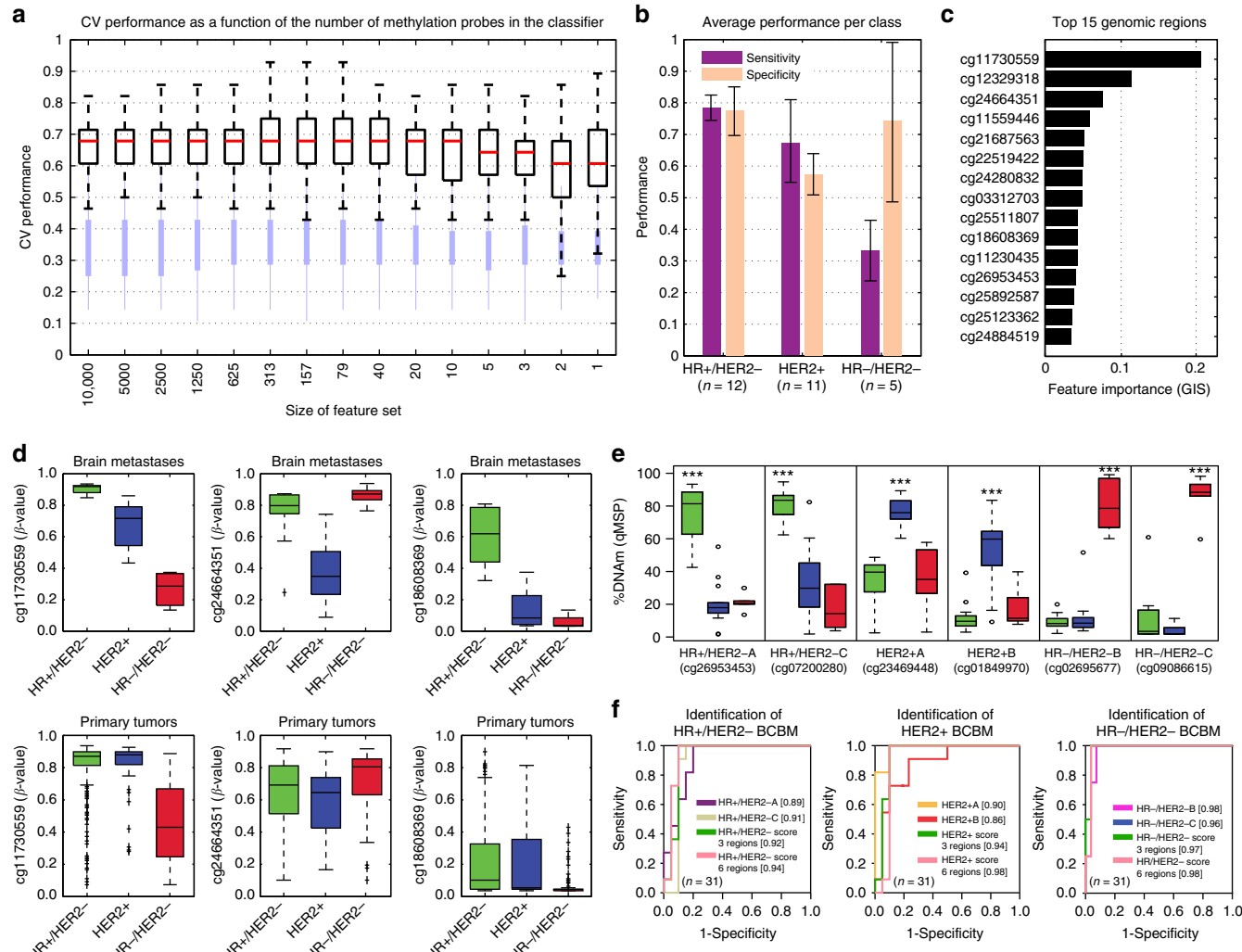

**Fig. 5** A DNA methylation-based classifier to predict BCBM subtypes. **a** Boxplots describing the CV performance across 100 repeats for RF classifiers to predict breast cancer subtypes. From left to right, decreasing numbers of features were used to construct the model (x-axis). The top and bottom of each box represent the first and third quartile, respectively; the internal red line represents the median values. Light blue bars depict the performance based on permuted class labels and represent the random background distribution. **b** Bar plots depicting the prediction performance as measured by sensitivity and specificity for each of the three subtypes. The bars show the average performance and interquartile range (error bars) across all models with 10 features or more across all repeats (HR+/HER2−; $n = 12$, HER2+; $n = 11$, and HR−/HER2−; $n = 5$). **c** Bar plots depicting the RF feature importance (mean decreases in Gini impurity score; GIS) of the 15 most predictive regions averaged across all models with 10 or more features and across all repeats. **d** Boxplots of DNAm levels ($\beta$-values) across our cohort stratified by subtypes (HR+/HER2−; $n = 12$, HER2+; $n = 11$, and HR−/HER2−; $n = 5$) and TCGA-BRCA cohort of primary breast tumors stratified to match our molecular subtype definitions (HR+/HER2−; $n = 443$, HER2+; $n = 83$, and HR−/HER2−; $n = 117$). Differences in the DNAm levels among the groups were statistically significant for all the cases (Kruskal–Wallis test; P-value < 0.0001). **e** DNAm levels assessed by qMSP for six genomic regions with differential DNAm among the three BCBM subtypes ($n = 31$). ***Wilcoxon test; P-value < 0.001. **f** ROC curves showing the prediction potential for the breast cancer subtypes ($n = 31$) for each of the six differentially methylated regions and combinations of three or six regions into BCBM molecular subtype-specific scores: HR+/HER2-score = DNAm level of HR+/HER2− minus DNAm level of HER2+ minus DNAm level of HR−/HER2; HER2+ score = DNAm level of HER2+ minus DNAm level of HR+/HER2− minus DNAm level of HR−/HER2; and HR−/HER2-score = DNAm level of HR−/HER2− minus DNAm level of HR+/HER2− minus DNAm level of HER2+; see Supplementary Data 10 for details about these regions. The AUC values are indicated between square brackets

These regions, alone or in combination, showed good discrimination potential for the respective BCBM subtypes using the HM450K data (AUC > 0.9; Supplementary Fig. 9a). We then designed qMSP assays for each region and evaluated their DNAm level in BCBM specimens ($n = 31$). Based on the qMSP analysis, each region was classified into poor, moderate, and good performance (genomic coordinates, nearby genes, primer sequences, and qMSP performance listed in Supplementary Data 10). Following qMSP evaluation with alternative primer sets, regions that still exhibited poor performance were evaluated using locus-specific bisulfite sequencing (Supplementary Fig. 9b; sequencing primer sequences listed in Supplementary Data 10). To take advantage of the quantitative nature and simplicity of the qMSP assays, we focused on regions with significant differential methylation among the BCBM subtypes (Wilcoxon test; P-value < 0.001; Fig. 5e) that, in addition, exhibited significant agreements with HM450K data (Spearman's $\rho$; P-value < 0.001; Supplementary Fig. 9c). Combining the DNAm levels of three or six regions

presented a slightly higher predictive potential (mean AUC = 0.95) than single regions (mean AUC = 0.91) in all cases (Fig. 5f). Thus, unlike the BM tumor of origin qMSP classifier, which required only one region per BM type, at least two genomic regions per BCBM subtype were needed for accurate BCBM classification (Fig. 5f). In agreement with the HM450K-based analysis of 126 regions, qMSP evaluation of these six regions identified the BCBM subtypes of four specimens lacking an IHC profile at the initial diagnosis (Supplementary Fig. 9d). Based on this evidence, this informative set of regions validated by qMSP was selected for the DNAm-based classification of BCBM subtypes (BrainMETH class C).

## Discussion

In this study, we constructed and validated DNAm classifiers to aid in the diagnosis of BMs. Routinely, the first step in the diagnosis of metastatic brain tumors is to exclude a possible primary CNS neoplasm[9,36,37]. Here, we found substantial differences in the DNAm landscapes of primary and metastatic brain tumors, allowing us to define DNAm signatures capable of distinguishing BM types and subtypes. These findings are in concordance with a pilot study performed by Euskirchen et al. using nanopore sequencing[38] and complement a recent study reported by Capper et al. using a large collection of DNAm profiles from primary CNS tumors[13]. Together, these data are indicative of the value of DNAm profiling for the comprehensive classification of primary and secondary tumors affecting the CNS. Of note, our study found that the combination of a small number of genomic regions provides high accuracy in discriminating primary from metastatic brain tumors. This is of great clinical and histopathological relevance, as we have demonstrated that these regions can be assessed by targeted qMSP using DNA from routine FFPE clinical specimens. Additionally, this approach can be easily adapted to cover other genomic regions of interest to increase the robustness and expand the applications of the DNAm classifier. However, we encountered limitations that were primarily related to the high density of CpG sites surrounding some of the informative genomic regions. For these regions, we

designed locus-specific bisulfite sequencing assays that successfully determined the DNAm status but failed to generate a non-binary quantification of the DNAm level. Therefore, we performed technical validation assays using regions with good and moderate qMSP performance. Regions capable of discriminating between GBM and BM by qMSP were included in the first step of the DNAm-based BM classifier (BrainMETH class A; summarized in Fig. 6a).

In order to obtain an accurate prognosis and apply tumor-specific therapies, the second step in the histopathological diagnosis for BMs is to identify the tissue of origin[6,39]. A recent study by Moran et al. provided strong evidence for the utility of DNAm profiling in determining the origin of cancer of unknown primary[12]. In this regard, we found that BCBM, LCBM, and MBM present intrinsic differences in DNAm profiles that reflect the features of the embryonic origin for each cell type. Thus, MBMs, which are derived from the neuroectodermal layer, showed substantial differences compared to the epithelial-derived BMs (BCBM and LCBM). Using the DNAm profiles and clinical-demographic data of our dataset, we tested for potential influences of prior therapeutic interventions. Our results suggested that prior treatments do not have a significant influence on the DNAm landscapes, at least of the regions selected for our study. We understand that, due to the range of multi-treatment approaches present in this BM cohort, these results are not definitive, and a study specifically designed to test the influence of various treatments is required to conclusively assess possible effects on BM DNA methylomes.

As building evidence indicates that DNAm signals are a useful diagnostic aid, we established a set of genomic regions using supervised learning to precisely classify BM according to the primary tumor of origin. In agreement with the presumptive histopathological diagnosis, BrainMETH accurately determined the origin of BM in patients with multiple primary tumors or with inconclusive anatomical pathologic evaluation. We found that these regions also exhibited good classification performance in identifying melanoma, breast, and lung primary tumors. This observation suggests that CpG sites with good discrimination potential for BM types may overlap genomic regions associated

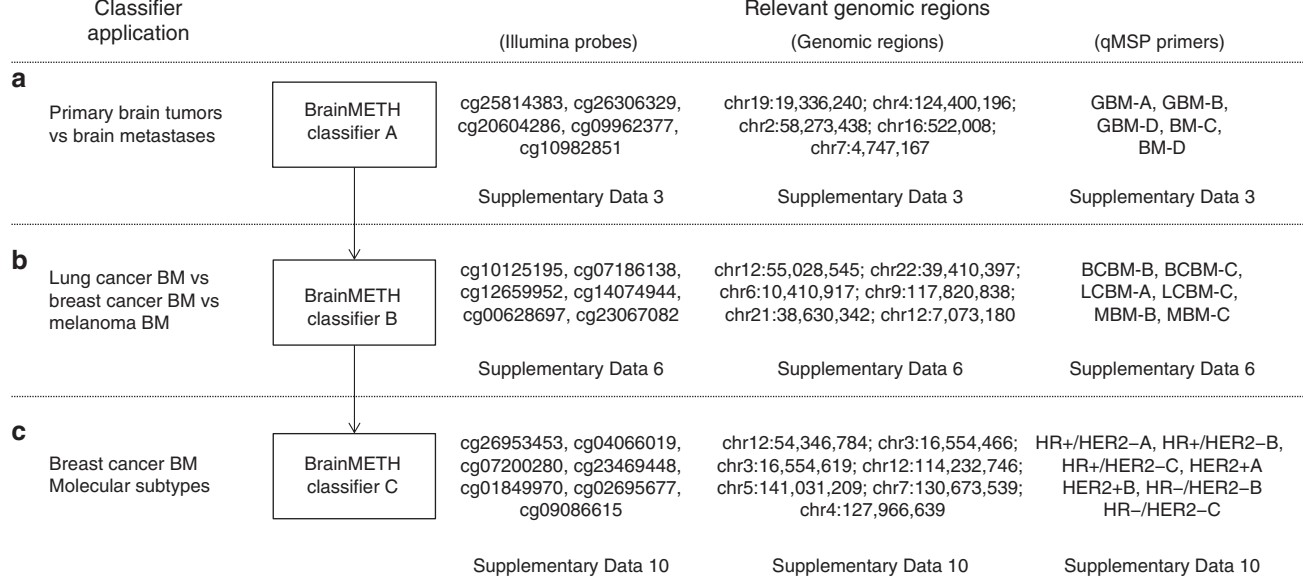

**Fig. 6** Summary of the BrainMETH classifiers. **a–c** BrainMETH classifiers designed to discriminate between primary and metastatic brain tumors (Classifier A), among BM from different tumor of origin (Classifier B), and among BCBM from different molecular subtypes (Classifier C). A set of relevant Illumina probes, genomic regions, and validated primer sets by qMSP is provided for each step of the BrainMETH classifier

with tumor-type-specific epigenomic signatures. Most importantly, we found that qMSP evaluations of three regions using DNA obtained from microdissected FFPE tissues showed a very good classification performance for BM specimens according to the tissue of origin. These regions were therefore selected for the second step of the BM DNAm classifier (BrainMETH class B; summarized in Fig. 6b).

Finally, once the diagnosis of the tumor of origin has been established, it is crucial to identify molecular features that can stratify patients and guide therapeutic decisions. Undoubtedly, the expression of HR and HER2 are the most meaningful predictive factors for the treatment of BCBM[40]. In clinical practice, therapeutically relevant subtypes are rarely evaluated in BCBM specimens and usually inferred from the IHC profiling of the primary tumor or extra-cranial metastases. However, significant discrepancies have been recently reported in the expression statuses of ER and HER2, the mutational burden, and gene expression profiles between matched primary and BCBM specimens[41–44]. Therefore, if feasible, it is currently recommended to perform a histopathological evaluation of the metastatic lesions to confirm the therapeutic subtype by at least reassessing the expression of HR and HER2[40]. To complement the conventional diagnosis, we explored the utility of DNAm profiling to classify BCBM into three therapeutically relevant subtypes (HR +/HER2−, HER2+, and HR−/HER2−). Overall, the BCBM subtype DNAm classifiers (BrainMETH class C) exhibited a substantially lower performance than the BM tissue of origin classifiers (BrainMETH class B). This difference may be influenced by the much larger difference in global DNAm patterns, and thus discriminatory power, between cancers from different tissues than between different breast cancer subtypes, in addition to the smaller sample size used to construct the BCBM subtype classifiers. Additional studies including new cases may improve the BCBM subtype classifiers, thereby enhancing their potential utility as a clinical application. In spite of this limitation, the BCBM subtype DNAm classifiers may still be useful as a supplement to the current histopathological diagnostic process. We identified a set of regions that accurately classified BCBM with initially unknown IHC profiling into clinically relevant subtypes. Since these classifiers only needed a small number of regions to achieve good accuracy, we conjecture that a qMSP assay for BCBM subtype determination may serve as a valuable tool in the clinical setting. We found that the combination of six regions twice that required for the BM tumor of origin BrainMETH classifier (Class B), showed a very good classification potential for each BCBM subtype. To facilitate the development of histomolecular applications of the BrainMETH, we provide a list of genomic regions with good classification performance for BCBM therapeutic subtypes that can be inexpensively evaluated using FFPE tissues (BrainMETH class C; summarized in Fig. 6c).

In summary, this study provides a comprehensive characterization of BrainMETH, a DNAm-based classifier of metastatic brain tumors. Data presented here further demonstrate the significant potential of DNAm profiling as a valuable molecular diagnostic tool, specifically for the diagnosis of intracranial metastases. IHC is a well-established, adaptable tool with a high success rate for the diagnosis of intracranial neoplasms. Although IHC remains the cornerstone of tumor diagnostics, we believe that the BrainMETH classifiers could serve as an effective ancillary tool in accurately diagnosing challenging cases; specifically, patients with occult primary tumors or poorly differentiated BM lesions. Furthermore, we believe that the quantitative and objective nature of the BrainMETH classifiers may help mitigate the level of inter-observer and intra-observer variability in IHC interpretation. Therefore, the BrainMETH classifiers, along with the recently reported primary CNS tumor DNAm classifiers[13] can

complement and assist the current diagnostic algorithm to improve objectivity in classifying different brain tumors. The BrainMETH classifiers assayed on DNA from FFPE tissues demonstrated good classification performance for BM originating from the three primary tumor types which most frequently metastasize to the brain. Further studies are required to assess the usefulness of DNAm classifiers in diagnosing patients with primary tumors with a less frequent incidence of BM, such as kidney, colorectal and ovarian cancers. Additional DNAm analyses using paired primary and BM specimens will continue to enhance our understanding of brain metastatic progression and identify novel prognostic and therapeutic applications. Despite this need for a more expansive assessment, the findings of this study represent the first steps in expanding the available tools for making an accurate diagnosis, which is crucial to determining prognosis and guiding therapeutic decisions in patients with BM.

## Methods

**Patients and tissue specimens processing**. In a multi-institutional effort, 165 patients with operable primary or metastatic brain tumors diagnosed at the Providence Saint John's Health Center (Santa Monica, USA), Melanoma Institute of Australia (Sydney, Australia), and the Swedish Medical Center (Seattle, USA) were enrolled for this study. All clinical-demographic data and patient-derived samples were collected under research protocols approved by the joint Institutional Review Board of Providence Saint John's Health Center/John Wayne Cancer Institute, the Western Institutional Review Board, Institutional Review Board of Swedish Medical Center, and the Sydney Local Health District (Royal Prince Alfred Hospital Zone) Human Ethics Review Committee. All patients signed an informed consent before joining the study. The experiments were performed in accordance with the World Medical Association Declaration of Helsinki and the National Institutes of Health Belmont Report. Tissues were de-identified and coded according to recommendations of the Health Insurance Portability and Accountability Act to ensure patient confidentiality.

**Histopathological evaluation and genomic DNA extraction**. Representative FFPE tissue block for each brain tumor specimen was selected by the pathologist from the Pathology Departments of the three institutions involved in the study. Neuropathologists reviewed tissue slides stained with hematoxylin & eosin for all specimens and identified areas with tumor cell enrichment (tumor purity) higher than 70%. After deparaffinization, hematoxylin staining was performed in serial tissue sections of 8 micrometerthick, and tumor tissues were needle microdissected from consecutive FFPE slides. Genomic DNA was isolated using ZR FFPE DNA MiniPrep (D3066; Zymo Research, Irvine, CA) following the manufacturer's instructions.

Ninety-six BM specimens from 94 patients with BCBM ($n = 30$), lung cancer (LCBM; $n = 18$), cutaneous melanoma (MBM; $n = 44$), and patients with both primary breast cancer and primary lung cancer ($n = 4$), were evaluated using the Illumina Infinium HumanMethylation 450K BeadChips (HM450K; Illumina Inc., San Diego, CA, USA). An extended description of the DNAm profiling experiments described here is available in our related data descriptor[45]. Briefly, one µg of genomic DNA was bisulfite converted using the EZ DNA Methylation-Direct Kit (D5021, Zymo Research Irvine, CA, USA). The efficiency of the bisulfite conversion was evaluated using the MethyLight assay for a panel of markers[46]. All the samples passing the quality control test were whole-genome amplified, enzymatically fragmented and repaired using the Infinium HD FFPE DNA Restore kit (WG-321-1002, Illumina Inc., San Diego, CA, USA). Finally, the fragmented and restored sodium bisulfite-modified DNA specimens were hybridized into the HM450K BeadChips and scanned using the Illumina iScan microarray scanner following the manufacturer's recommendations (Illumina Inc., San Diego, CA, USA).

**Quantitative-methylation-specific PCR**. Sodium bisulfite modification was performed on 200–500 nanograms of genomic DNA using EZ DNA Methylation-Direct (D5021, Zymo Research, Irvine, CA, USA) following manufacturer recommendations. Target DNAm of GBM and BM genomic regions was performed using primers sets described in Supplementary Data 3, 6, and 10. The quantitative amplification of methylated and unmethylated alleles was performed in CFX96 Touch™ Real-Time PCR detection system (185–5196; Bio-Rad Laboratories, Irvine, CA, USA), and the $\Delta Ct$ ($\Delta Ct$ = mean Ct methylated – mean Ct unmethylated) was calculated for each CpG site. The relative DNAm level was established by using the $2{-}\Delta Ct$ method. The percentage of DNAm was estimated by using a logarithmic equation derived from the analysis of a standard curve of serial dilutions of the universal methylated control (D5014, Zymo Research, Irvine, CA, USA) in universal unmethylated control (D5014, Zymo Research, Irvine, CA, USA) for each genomic region. Unless otherwise indicated in the text, BM type or subtype-specific DNAm scores were established as the average DNAm level for the BM type or subtype-specific region(s) minus the average DNAm level for the contrasting BM type or subtype-specific region(s).

**Locus-specific bisulfite sequencing**. Ten genomic regions with poor qMSP performance were amplified with sodium bisulfite conversion-dependent primers (Supplementary Data 3, 6, and 10). After amplification of methylated and unmethylated universal controls (D5011-1 and D5014-1, respectively, Zymo Research, Irvine, CA, USA), PCR products were purified with the QIAquick PCR Purification Kit (28106, Qiagen, Hilden, Germany) and subsequently verified in a 2.2% agarose Flashgel (57032, Lonza, Rockland, ME, USA). Successfully amplified samples were then quantified by UV absorption spectrophotometry and sequenced using internal sequencing primers (Supplementary Data 3, 6, and 10) using the Eurofins MWG Operon PlateSeq service (Eurofins Genomics, Louisville, KY). Sequencing results were then visualized using Chromas Lite v2.6.5 (Technelysium, Brisbane, Australia).

**Histopathological and IHC evaluation of BCBM specimens**. The BCBM specimens were evaluated at the CLIA-certified Department of Pathology, Providence Saint John's Health Center, accredited by the College of American Pathologists (CAP). The BCBM samples were classified into molecular subtypes according to the expression levels of ER and PgR by IHC. HER2 was assessed by IHC and/or in situ hybridization (ISH) assays. The FFPE tissue slides were sectioned at 4 micrometer thick and mounted on plus-coated glass slides, and immunohistochemically stained using a Ventana BenchMark ULTRA automated slide stainer (Roche Diagnostics, Indianapolis, IN, USA). Antibodies used were anti-Estrogen Receptor (SP1, dilution 1:1000, #790-4324, Ventana Medical Systems, Tucson, AZ, USA), anti-Progesterone Receptor (1E2, dilution 1:1000, #790-2223, Ventana Medical Systems, Tucson, AZ, USA) and PATHWAY anti-HER-2/neu (4B5, dilution 1:166, #790-2991, Ventana Medical Systems, Tucson, AZ, USA). The scoring criteria for these markers were based on the ASCO/CAP guidelines[47,48]. Briefly, ER and PgR were considered positive if there was staining of the nucleus in at least ≥ 1% of the tumor cells in the sample. HER2 was considered positive for IHC 3+ or ISH amplified if single-probe average HER2 copy number > 6.0 signals/cell or dual-probe HER2/CEP17 ratio ≥ 2.0. BCBM specimens were grouped according to the expression of these routinely clinically evaluated markers into a-HR+/HER2−, b- HR any/HER2+ (HER2+), and c- HR−/HER2−.

**Access to NCBI GEO DNA methylation datasets**. HM450K data generated from 100 normal brain tissues; including 25 frontal cortex, 25 superior temporal gyrus, 25 entorhinal cortex, and 25 cerebellum specimens (GSE43414). Additionally, DNA methylomes generated from 152 GBM specimens (GSE85539) and 59 LGG specimens from patients enrolled in the phase III study EORTC 26951 (GSE48461). These data were integrated with the DNA methylomes from 16 melanoma BM (GSE44661)[15,16]. These datasets were accessed using the R/Bioconductor GEOquery package v2.46.13, normalized using the SWAN method[49] on the R/Bioconductor wateRmelon package v1.19.1, and annotated using the R/Bioconductor FDb.InfiniumMethylation.hg19 package v2.2.0. DNAm data (.idat files) from 2085 primary CNS tumors (GSE90496) were combined and normalized with DNAm data from our BM cohort using the Noob background correction (processNoob) function in the R/Bioconductor minfi package v1.18.4.

**Access to TCGA data and classification of patients**. Clinical data for GBM, breast cancer, lung cancer, and cutaneous melanoma patients were downloaded from the Broad GDAC Firehose website (https://gdac.broadinstitute.org/) in April 2017. Genome-wide DNAm data for all these patients were retrieved from the National Cancer Institute Genomic Data Commons Portal (https://gdc.cancer.gov/) using the R/Bioconductor TCGAbiolinks package v1.2.5[50]. We selected 60 patients with GBM from the TCGA-GBM project with an absolute tumor purity higher than 70% and genome-wide DNAm level assessed by the HM450K platform. This cohort included patients with demographic characteristics paired with the cohort of patients with BM and a representative sampling of the GBM molecular subtypes. Of these cases, 7% had a high glioma CpG island methylation phenotype (G-CIMP), which is similar to the reported clinical frequency of G-CIMP[25]. We selected 380 primary lung cancer specimens corresponding to adenocarcinoma (n = 136) and squamous (n = 171) from the TCGA-LUAD and TCGA-LUSC projects, respectively. From the combined list of TCGA lung cancer patients (n = 1,026), we excluded specimens obtained from patients with Stage IV (n = 32), unknown stage (n = 3), T4 from the American Joint Committee on Cancer (AJCC) 5th or 6th edition (n = 27; since these cases can belong to the category of malignant pleural effusion currently considered M1a on the AJCC 7th edition), recurrent sites (n = 2), unavailable location (n = 2), and tumor purity < 70% (n = 580). Additionally, we selected 83 primary cutaneous melanoma specimens from 466 cases included in the TCGA-SKCM project. Our exclusion criteria included specimens obtained from regional cutaneous or subcutaneous lesions including satellite and in-transit metastases (n = 75), regional lymph node metastases (n = 220), distant organ metastases (n = 67), unavailable location (n = 2), primary melanoma lesions from patients with stage IV (n = 3), unavailable AJCC 7th edition stage (n = 2), and tumor purity < 70% (n = 16). We identified and classified primary breast cancer specimens from the TCGA project with complete IHC information (n = 914) and HM450K evaluation into 1-HR +/HER2− (n = 443), 2-HRany/HER2+ (HER2+, n = 83), and 3-HR−/HER2−

(n = 117). Briefly, from the 1105 breast cancer samples included in the TCGA-BRCA database, we excluded specimens obtained from metastatic sites (n = 7), male patients (n = 12), unavailable gender information (n = 2), and unavailable or indeterminate ER, PgR or HER2 statuses (n = 170). Cases with discordant results for HER2 expression by IHC and ISH were individually evaluated using the information provided in the pathological reports, and classified according to ASCO/CAP guidelines (n = 15)[48].

**Statistical and bioinformatics analyses**. Tumor purity of all TCGA samples was assessed using the CPE method[26]. Differentially methylated genomic regions among the groups, as well as a metric of the variance explained by each group, were identified using the F-Ratio and F-Distribution based on the hypothesis test for the one-way ANOVA. All P-values were two-sided and corrected for multiple comparisons using either Bonferroni or the FDR correction methods, as indicated in each case. DNAm data were dichotomized into methylated (β-value ≥ 0.9) and unmethylated (β-value ≤ 0.1) categories. Receiver operating curves (ROC) were used to estimate the sensitivity and specificity of the brain tumor classification method. The AUC was calculated for each ROC to evaluate the accuracy of brain tumor classification based on DNAm observed in genomic regions. The gene ontology enrichment analyses for differentially methylated genomic regions were performed using the Genomic Regions Enrichment of Annotations Tool (GREAT) v3.0.0 with the basal plus extension association rule considering proximal (up to 5 kb) as well as distal (up to 1 Mb) genes, only including curated regulatory domains[27]. HCL analyses were performed using Euclidean distance and average linkage clustering to identify relationships between genomic regions and brain tumors. PCA was used to evaluate the overall variability, identify possibly correlated variables, and to visualize the distance between selected brain tumor DNAm profiles. The robustness of phylogenetic tree reconstruction was determined by bootstrap resampling using 1,000 iterations using HCL analyses. The HCL, PCA, terrain maps, and bootstraps analyses were performed using the MultiExperiment Viewer v4.9[51]. Phylogenic trees were visualized using the FigTree Viewer v1.4.3. The t-SNE technique[52,53] and the UCSC TumorMap visualization tool[54] were employed to generate unsupervised clustering of the entire BM cohort (n = 96) along with a reference cohort of primary CNS tumors (GSE90496, n = 2085). The t-SNE was generated using the Rtsne R package v0.13 with theta = 0 for exact t-SNE and the TumorMap was generated using the online application (https://tumormap.ucsc.edu/) with the standard configuration. The multivariate DCA was used to find the most relevant genomic regions whose DNAm level was associated with each BCBM molecular subtype. DCA scores were established using the decorana function contained in the R vegan package v2.3.5 and visualized using the COA function on the MultiExperiment Viewer v4.9[51]. The nearest shrunken centroids (NSCs) algorithm (initial parameters: delta > 5, minimum correlation = 0.5, number of bins = 24) was applied to identify genomic regions that more accurately predict BCBM molecular subtypes. The NSCs were computed using the function pamr.train contained in the R pamr package v1.55. DNAm classifiers to predict tumor of origin and BCBM molecular subtypes were trained and tested using the RF algorithm[32] applying the RF implementation for MATLAB v0.02 downloaded from http://code.google.com/p/randomforest-matlab/. The RF classification models were run with 5000 trees each, default settings for the other parameters, and were trained and tested using a three-fold stratified CV strategy. Using only the samples in the training set, we first selected the 10,000 genomic regions which showed the largest variation in the median β-value across the different classes to train an RF model. Next, we selected the half of the genomic regions with the largest feature importance scores and retrained the RF using only this half of the regions. This process was iterated in 15 incremental steps, approximately halving the size of the feature set at each step, resulting in the 15th iteration in an RF classifier trained on only one genomic region. Each of the 15 RF classifiers was evaluated using the samples in the test set. This iterative backward elimination procedure was repeated using each of the three folds as the test set. Note that the test samples were never used for training. The whole procedure was replicated 100 times. The same procedure was repeated with permuted class labels for each of the 100 repeats in order to build a distribution of the performance of a random classifier. The feature importance was measured as the Gini impurity score used for the calculation of splits in the decision tree during training of the RF. Specifically, the reported feature importance scores are the mean decreases in the Gini impurity score. This is the standard method to calculate and report feature importance metrics in RF.

## Data availability
All HM450K raw and normalized data that support the findings of this study have been deposited in the NCBI's Gene Expression Omnibus (GEO), and datasets are accessible through the series records GSE108576 and GSE44661. A detailed explanation of these datasets is available in our related data descriptor[45]. All other data that support the findings of this study are available from the corresponding author upon request.

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

## Acknowledgements

We are grateful to Dr. Ian Hutchinson for his critical revision of the manuscript. This work was supported by the National Cancer Institute, National Institutes of Health (#R01CA167967 to D.S.B.H.; #P01CA077852 Core B to I.S. and T.A.K.; and #U24CA210952 to T.A.K.); the Dr. Miriam and Sheldon G. Adelson Medical Research Foundation (to D.S.B.H.); the Ben and Catherine Ivy Foundation (to P.H. and C.S.C.); the AVON Foundation Breast Cancer Crusade (#02-2015-061 to D.S.B.H. and D.M.M.); the Associates for Breast and Prostate Cancer Studies (ABCs) award (#88737700140000 to J.I.J.O. and D.M.M.); the Fashion Footwear Association of New York (FFANY)

foundation award (#88737890550000 to J.I.J.O. and D.M.M.); and the John Wayne Cancer Institute Translational Research Fund (to M.P.S. and D.M.M.).

## Author contributions

J.I.J.O., T.A.K., D.S.B.H. and D.M.M conceptualized the study. J.R.J., Y.T. and M.E.B. performed the histopathological evaluation of tissue samples. G.B., J.F.T., G.V.L., C.S.C. and D.F.K. performed the selection of patients and clinical data annotation. J.S.W., P.H., and X.W. contributed with the tissue logistics and annotation of clinical data. D.M.M. designed the wet lab experiments. J.I.J.O. and A.O.M.-P. performed the wet lab experiments. T.A.K., M.P.S., I.S. and D.M.M. performed the data normalization, statistical evaluations, and bioinformatics analyses. D.S.B.H. and D.M.M. provided general guidance and oversaw the study. J.I.J.O., T.A.K., M.P.S., G.B., J.R.J., G.V.L., R.A.S., D.S.B. H. and D.M.M. interpreted the results. J.I.J.O., T.A.K., A.O.M.-P. and D.M.M. wrote the manuscript. All authors read and approved the final manuscript before submission.

## Additional information

**Competing interests:** The authors declare no competing interests.

