## [Peer Review File · Nature Communications]

Reviewer #1 (Remarks to the Author):

In their manuscript titled, "Epigenetic profiling for the molecular classification of metastatic brain tumors," Orozco et al, utilize Infinium DNA methylation microarray data to identify CpG methylation patterns and signatures capable of distinguishing: i) GBM from brain mets; ii) lung, breast, and melanoma derived brain mets from each other; and iii) Breast cancer subtypes from breast cancer brain mets. The work is carried out well from a technical stand point, and provides a nice demonstration that DNA methylation patterns can be used to discriminate tissue of origin and breast cancer subtype of brain metastases. I have the following constructive criticisms:

- 1) It is possible that prior treatments may modify the methylation patterns, including those used in the three different classifiers that are presented. The Supplementary Data does not present information on the prior treatments for most of the specimens. Also, it is possible that cancers that recur after primary treatment, particularly with radiation therapy or chemotherapy, may have different properties than metastases that are found at the time of presentation. The authors should examine whether such parameters alter their conclusions, and to what extent.
- 2) The Supplementary Data 1 material are useful. However, in addition to this, it would be useful to summarize the information by various clinicopathological parameters in a table format.
- 3) While this study was in review, a report from Capper et al, Nature, 2018, reported the development of a DNA methylation based classification of central nervous system cancers. The authors should contextualize their work with this report.
- 4) It would be helpful to have a more careful assessment of how often the classification panels they have developed would be useful. What fraction of cases are not currently identifiable by standard-of-care histopathological and clinical analysis? What percentage of these would it be possible to make these DNA methylation measurements? A candid evaluation of the true utility of these predictors would be important in terms of impact.

Reviewer #2 (Remarks to the Author):

In the manuscript entitled "Epigenetic profiling for the molecular classification of metastatic brain tumors" Orozco and colleagues used DNA methylation array data to identify subsets of CpG dinucleotides whose methylation profiles for classification of primary gliomas and brain metastases from lung, breast and melanoma primary tumors. Overall this is an interesting study and has important implications with respect to identification of tumor of origin, diagnostic and treatment

schemes. While previous reports have identified tissue of origin determinations of gliomas and other primary tumors using DNA methylation data, this report is novel in that it compared primary brain tumor data to brain metastases.

Clarification of the questions below would help to improve the manuscript:

1. The CpG island methylator phenotype (CIMP) is an important classifier of low and high grade gliomas (G-CIMP), and has been described in melanomas and breast tumors by TCGA and others. While G-CIMP represents a minority of GBM patients, the DNA methylation profiles of G-CIMP patients are striking compared to non-G-CIMP patients. How has G-CIMP status specifically been addressed in this study? It seems that G-CIMP-specific markers and tumors should be removed from this analysis and this step should be included in the HM450K probe filtering steps for all tumor types.

2. The qMSP data and discussion indicate that qMSP assays were classified into poor, moderate and good performance. The poor performing assays were not utilized, however, does this point to the failure of these assays to validate the array data? Were other qMSP assays attempted? Would other technologies, such as bisulfite sequencing, be a better and more effective means of validation?

3. The description of performance using differing numbers of markers is shown in figures 3a and 5a. In Figure 3a, it seems that a panel of >20 markers is sufficient for accurate classification, and in Figure 5a, one could argue that as little as three markers are sufficient. How were these features selected? In Figure 5a, it seems that only one marker, cg11730559, outperformed all other markers in the analyses. Were larger panels built upon this initial marker? How were the levels of “feature importance” quantitatively measured? Is this a composite of specificity and sensitivity?

4. No specific genes were listed in the figures – only Illumina cg probe IDs. What are the gene regions identified from these analyses? Are these important for the biology of these tumors? What is the consequence of differential DNA methylation in terms of gene regulation? While gene expression aspects are not critical for identifying DNA methylation-based classifiers, an analysis of gene expression may help identify pathways for treatment or other diagnostic means.

5. In Figure 2e, why were only a sub-cohort of women used in this analysis?

Reviewer #3 (Remarks to the Author):

The current work by Javier Orozco et al. generated epigenetic profiling for the molecular classification of metastatic brain tumors, and particularly they constructed the DNAm-based brain classifier (BrainMETH) which has significant value in the molecular diagnosis of intracranial metastases.

While brain metastasis-defining epigenetic alterations which can be utilized to further develop DNA methylation profiling as a critical tool in the histomolecular stratification of patients with brain metastases, the work suffers from many limitations, per below.

Major concerns:

1. Figure 1: to identify intrinsic differences in the epigenetic profiles of metastatic brain tumors, the author generated DNAm signatures for 96 microdissected BM tissues. In the experiment, the author try to decrease potential biases associated with the gender of BM patients, they excluded 105,422 probes recognizing regions located in sex chromosomes or with proven cross-reactivity with sex chromosomes. This reviewer doesn't agree with conclusions. We think there are no need to exclude the probe located in sex chromosomes, because the majority breast cancer patients are female, and sometimes the prognosis/ treatment response are very different among female and male lung cancer brain metastasis patients. Furthermore, ratios between male and female patients in every subtype of breast cancer or lung cancer are drastically different, therefore in these cases the gender influence could not be excluded. Lastly, Fig.1A simply describes which genomic regions were excluded from the analysis is irrelevant or difficult to unambiguously analyze. This is not a result, it is a technical issue and should be presented in the methods part only.

2. Figure 3: authors identified the genomic regions difference among breast/ lung/melanoma brain metastasis for the prediction of the tumor of origin. However, did authors verify genomic region differences between the brain metastasis with primary sites (breast cancer, lung cancer and melanoma cancer) using TCGA data, plus their own patient data? Second (Fig.3A), classifiers to predict tumor of origin for BM were identified among DNA methylation patterns. Using data set training and testing, this yielded fantastic results, both in terms of sensitivity and specificity. It also showed that using combination of 40 or more (up to 10,000) classifiers a maximal (approximately 98%) prediction performance is achieved. In light of these results, Fig.5A is hard to understand. Here, classifiers to predict BCBM molecular subtypes are derived and tested with the same approach as in Fig.3A. Results showed that, on average, the predictive power of one classifier alone is the same as the combination of 3, 5, 10, up to 10,000. This is very confusing and an explanation for this phenomenon should have been provided in the text. Furthermore, the predictive power is much less

striking, range 30% to 80%. Is this still valuable clinically? What is the difference between Fig.5B and Fig.5F? Third, (Fig.3D), genomic regions which predicted well BM tissue origin yielded very different results when applied to primary tumors of the same tissues, e.g., breast cancer left and right panels, lung right panel). It is conceivable that BM of breast cancer have unique features compared to primary breast cancer. However, this result and the potential impact on patient-treatment decisions were not discussed.

3. Figure 4: authors used the DNAm profiling to predict BCBM molecular subtypes. However, did authors analyze the DNAm profiling to predict lung cancer metastasis/melanoma cancer brain metastasis subtype?

4. Figures 5 & 6: authors found the breast cancer brain metastasis molecular subtype-specific DNA methylation profiles and used them to aid in the identification of breast cancer brain metastasis molecular subtype. Are these DNA methylation signatures helpful to distinguish different type of primary breast cancer type? Or differences of DNAm profiles among different subtypes of primary breast cancer types are similar to differences in DNA profiles among different three BCBM subtypes? Further (Fig.5D), same concerns are raised as in Fig.3D: genomic regions which worked to predict BM molecular subtype yielded drastically distinct results when applied to primary tumors. Accordingly, is using these regions for molecular subtype determinations still valid and applicable?

5. Did authors ever compared the DNAm profiling between GBM and three different brain metastasis cancer type separately?

Minor concerns:

1. The bar graphs for feature importance in Fig.3C and Fig.5C do not have scale numbers.

2. Authors are recommended to discuss further the use of results achieved, e.g. the refinement of BM classification, correlations to aggressiveness, stage of disease, etc.

Reviewer #1 (Remarks to the Author):

In their manuscript titled, "Epigenetic profiling for the molecular classification of metastatic brain tumors," Orozco JI et al, utilize Infinium DNA methylation microarray data to identify CpG methylation patterns and signatures capable of distinguishing: i) GBM from brain mets; ii) lung, breast, and melanoma derived brain mets from each other; and iii) Breast cancer subtypes from breast cancer brain mets. The work is carried out well from a technical stand point, and provides a nice demonstration that DNA methylation patterns can be used to discriminate tissue of origin and breast cancer subtype of brain metastases.

I have the following constructive criticisms:

1) It is possible that prior treatments may modify the methylation patterns, including those used in the three different classifiers that are presented. The Supplementary Data does not present information on the prior treatments for most of the specimens. Also, it is possible that cancers that recur after primary treatment, particularly with radiation therapy or chemotherapy, may have different properties than metastases that are found at the time of presentation. The authors should examine whether such parameters alter their conclusions, and to what extent.

Response to Reviewer: Following this suggestion, we have reviewed prior treatment approaches for all patients in this study and have included this information, as well as all the previously presented clinical and demographic data, in a new **Supplementary Table 1**. While, as expected, we found that most of the patients with metastatic brain tumors had received either systemic therapy and/or radiation therapy previous to the brain metastasis (BM) surgery, we were able to identify patients with treatment-naïve metastatic brain tumors (n=28) that allowed us to perform the proposed DNA methylation comparative analyses.

First, we categorized prior BM disease-specific treatment approaches into the following sub-groups: **A- Prior systemic therapy** (n=55 patients) includes any type of adjuvant or neoadjuvant drug-based therapy for the primary tumor, and also any drug-based therapy for metastatic disease before the metastatic brain tumor specimen was removed. We further refined the systemic treatment group according to the type of systemic therapy into **A1- Prior chemotherapy** (n=50 patients) includes the use of any type of chemotherapeutic agent to treat the primary tumor, extracranial metastases, and/or brain metastases before the removal of the analyzed specimen; and **A2- Prior targeted therapy** (n=38 patients) includes the use of any type of targeted agent in the neoadjuvant, adjuvant and/or metastatic settings. In our cohort of patients, we identified the use of anti-HER2 therapy, endocrine therapy, anti-EGFR therapy, BRAF inhibitors, immunotherapy, or other investigational targeted drugs, such as trastuzumab, lapatinib, tamoxifen, aromatase inhibitors, erlotinib, afatinib, dabrafenib, vemurafenib, ipilimumab, pembrolizumab or nivolumab (detailed in

the new **Supplementary Table 1**). **B- Prior radiation therapy** (n=36 patients) includes those cases in which radiation therapy was used for the loco-regional treatment of the primary tumor, extracranial metastases, and/or the use of stereotactic radiosurgery or whole brain radiotherapy before the brain metastasis was surgically removed and included in our study.

Second, we investigated the potential impact of the treatment modalities on the generation of additional BM DNA methylation subsets. This evaluation was performed by exploring the distribution of BM specimens on the principal component analyses (PCA), based on the DNA methylation levels of the selected set of genomic regions with the potential to identify the origin of each BM specimen (31,818 genomic regions as shown in **Fig. 2c** and **Supplementary Table 4**). Due to the variable, disease-specific treatment modalities, these evaluations were performed independently for each type of brain metastasis. The multidimensional scaling analyses showed that prior treatment modalities did not generate additional BM subsets (new **Supplementary Fig. 5**). In fact, we found a negligible number of genomic regions with significant treatment related differential methylation levels (<0.2% of the 31,818 evaluated genomic regions; **Pages 11-12** and new **Supplementary Fig. 5**).

We understand that, because our study was not designed to specifically carry out this comparison, these results are not conclusive and would require an independent study designed to definitively address this interesting question. We have therefore indicated this limitation in the Discussion section (**Page 21**).

2) The Supplementary Data 1 material are useful. However, in addition to this, it would be useful to summarize the information by various clinicopathological parameters in a table format.

Response to Reviewer: Thank you for indicating this, as mentioned above, the complete information provided in **Supplementary Data 1**, as well as the information related to therapeutic approaches, have been organized in a tabular format and presented as a new **Supplementary Table 1**.

3) While this study was in review, a report from Capper D et al, Nature, 2018, reported the development of a DNA methylation based classification of central nervous system cancers. The authors should contextualize their work with this report.

Response to Reviewer: Our DNA methylation-based classification of metastatic brain tumors complements the study recently published by Capper D et al in Nature¹, in which they employed DNA methylation data to classify primary central nervous system (CNS) tumors, and emphasizes the value of epigenetic profiling to better stratify tumor tissues. Our study additionally provides the necessary details to select and validate a small number of highly informative genomic regions, which can be used as a cost effective profiling alternative to the Illumina Human Methylation 450K

microarray technology. In our revised version, we have discussed the relevance of these findings for the development of histomolecular diagnosis, and the importance of considering both studies for a more comprehensive classification of primary and metastatic CNS tumors (**Page 20**).

As a reference, here we provide a visualization of our BM samples alongside the CNS tumor specimens included in Capper D et al. using data dimensionality reduction techniques. To decrease potential biases in the alternative DNA methylation data normalization approaches, we combined the raw .idat files for the vast majority of samples used in Capper et al. 2018 with the raw .idat files from our BM samples, and normalized them using the processNoob function in the *minfi* Bioconductor package. For this analysis we used the set of 14,494 probes that were identified as informative in this study, as well as subsets of the most variable genomic regions (n=1,000, n=100, and n=50). For alternative visual comparisons, we used both the t-distributed stochastic neighbor embedding (t-SNE; **left plot**), the same technique used by Capper D et al. 2018, and the TumorMap visualization tool (**right plot**), which in addition to combining multidimensional scaling approaches, further enhances connections between specimens by using a spatial correlation analysis (SCA)². These multidimensionality reduction analyses resulted in a clear clustering of the brain metastasis specimens within a larger context of other CNS tumor types and demonstrate that metastatic brain tumors have unique DNA methylation signatures that can be further refined for pathological stratification. This information has been added in the Results section (**Page 8** and **Supplementary fig. 1b-c**) and Discussion section (**Page 20**).

*This figure includes the multidimensional data reduction techniques presented in the new Supplementary Figure 1c and 1d. Briefly, these plots were generated following the reviewer's suggestion to contextualize our study based on metastatic brain tumors in the recently reported dataset by Capper D et al., Nature 2018 including a representation of a wide variety of primary central nervous system (CNS) tumors. To visualize the relationships based on DNA methylation profiles between metastatic brain tumors with other types of intracranial tumors we employed the t-distributed stochastic neighbor embedding (t-SNE; **left plot**), the same technique used by Capper D et al., and the TumorMap tool (**right plot**), which in addition to combining multidimensional scaling approaches, further enhances connections between specimens by using a spatial correlation analysis (SCA).*

4) *It would be helpful to have a more careful assessment of how often the classification panels they have developed would be useful. What fraction of cases are not currently identifiable by standard-of-care histopathological and clinical analysis? What percentage of these would it be possible to make these DNA methylation measurements? A candid evaluation of the true utility of these predictors would be important in terms of impact.*

Response to Reviewer: Currently, the differential diagnosis of brain tumors can be achieved with clinical information, neuro-imaging data, and histopathological features (BM usually resembles the histology of primary tumor) for the majority of cases. In addition to the clinical and histological features, a panel of immunohistochemistry (IHC) markers is required to assist in the accurate identification of tissue of origin of metastatic brain tumors. However, there are still cases that may be challenging to diagnose, like undifferentiated tumors or in BM from cancer of unknown primary (CUP). Brain metastases from CUP represent 15-25% of all BMs. Nevertheless, the histological and IHC evaluation of CUP cannot determine the primary tumor in up to 75% of cases³. In addition, inter-observer and intra-observer variation in IHC evaluation can lead to discrepancies in IHC evaluation. We believe that the *BrainMETH* classifiers constructed and validated by quantitative methylation-specific PCR in our study have the potential to assist in the anatomic-pathological evaluation of metastatic brain tumors for cases which are challenging to diagnose, as well as complement the current diagnostic process for BM. Furthermore, these classifiers are able to identify the specific molecular subtypes of patients with BCBM, which may greatly impact the clinical management of these patients. *BrainMETH* classifiers were trained and tested in specimens from patients with the three most frequent types of brain metastases (up to 90% of the reported cases). Therefore, further studies are required to expand its application to other less frequent types of brain metastases. Complementary to this, DNA methylation profiling has been shown to be a robust and reproducible method in small samples from archival tissue. Therefore, we assert that prospectively collected, as well as archived FFPE tumor tissues could be evaluated by the *BrainMETH* classifiers. We have discussed this observation in our revised manuscript (**Page 23-24**).

Reviewer #2 (Remarks to the Author):

In the manuscript entitled "Epigenetic profiling for the molecular classification of metastatic brain tumors" Orozco and colleagues used DNA methylation array data to identify subsets of CpG dinucleotides whose methylation profiles for classification of primary gliomas and brain metastases from lung, breast and melanoma primary tumors. Overall this is an interesting study and has important implications with respect to identification of tumor of origin, diagnostic and treatment schemes. While previous reports have identified tissue of origin determinations of gliomas and other primary tumors using DNA methylation data, this report is novel in that it compared primary brain tumor data to brain metastases.

Clarification of the questions below would help to improve the manuscript:

1. The CpG island methylator phenotype (CIMP) is an important classifier of low and high grade gliomas (G-CIMP), and has been described in melanomas and breast tumors by TCGA and others. While G-CIMP represents a minority of GBM patients, the DNA methylation profiles of G-CIMP patients are striking compared to non-G-CIMP patients. How has G-CIMP status specifically been addressed in this study? It seems that G-CIMP-specific markers and tumors should be removed from this analysis and this step should be included in the HM450K probe filtering steps for all tumor types.

Response to Reviewer: This is an interesting observation and we agree that the glioma CpG island methylator phenotype (G-CIMP) is perhaps one of the most relevant clinical applications of DNA methylation profiling. We also agree that G-CIMP neoplasms present not only a distinct DNA methylation profile but also substantial clinical, pathological and prognostic differences with non-G-CIMP neoplasms. Thus, during our original selection of the TCGA GBM patients, we included a percentage of G-CIMP cases similar to the clinical frequency of these cases (7%). We have now included this information in the revised manuscript (**Page 29**). Additionally, to investigate the reviewer's observation about the potential influence of the G-CIMP on the performance of the *BrainMETH* classifiers, we have performed two evaluations. First, based on the most recent comprehensive epigenetic evaluation of GBM, we identified 131 HM450K probes that were specifically associated with high G-CIMP (see Figure 3A on Ceccarelli et al. *Cell* 2016)⁴. Of note, we found that 95 of these G-CIMP-associated genomic regions were part of our initial CpG dataset (n=211,552 genomic regions). Only one of these probes (cg06821120) was significantly hypermethylated in GBM specimens when compared to brain metastases and, most importantly, none of them were retained in the **BrainMETH classifier A**. Second, to assess the potential influence of G-CIMP GBM cases in the comparison with BM specimens, we repeated the multidimensional scaling and included **(a)** - differential coloring of high G-CIMP and non-G-CIMP cases and **(b)** - exclusion of the high G-CIMP cases from the analysis. As represented below, high G-CIMP cases were not included in specific subsets on the three-dimensional principal component analysis (PCA) space (**left PCA plot**). The exclusion of these cases did not significantly affect the

multidimensional reduction analysis performance (**right PCA plot**). Including all the GBM specimens, we found that PCA explained a mean of 84.14 ± 0.6 percent of the cumulative variance (as described in **Page 8**). Excluding G-CIMP GBM specimens, we found that PCA explained a mean of 84.16 ± 0.4 percent of the cumulative variance. Thus, in view of the negligible influence of the G-CIMP-associated genomic regions on the performance of the *BrainMETH* classifiers, we think that the existence of epigenetic variability in the GBM cohort contributes to the development of a more robust classifier.

The principal component analyses presented in this figure show the spatial distribution of the glioblastoma (GBM) and the brain metastasis (BM) specimens included in the study and presented in Figure 1c. **The left plot** contains all the GBM cases and the cases identified as G-CIMP are highlighted in yellow. **The right plot** represents the resulting PCA after excluding the cases with G-CIMP.

2. The qMSP data and discussion indicate that qMSP assays were classified into poor, moderate and good performance. The poor performing assays were not utilized, however, does this point to the failure of these assays to validate the array data? Were other qMSP assays attempted? Would other technologies, such as bisulfite sequencing, be a better and more effective means of validation?

Response to Reviewer: Based on the DNA methylation microarray data, the regions identified and described in each *BrainMETH* classifier are highly informative and could be adapted to different DNA methylation screening approaches. In our validation phase, we decided to employ qMSP because of its quantitative nature and its potential cost-effective adaptation to different laboratory settings. However, as with other DNA methylation-based approaches, this technique has important limitations, including but not limited to, CpG density that diminishes the ability to design efficient primers and increased sequence redundancy after sodium bisulfite conversion. During the initial quality control evaluation of each region, we employed additional qMSP primer sets and combinations for regions showing poor performance. Yet, independently of the additional primers

and several PCR conditions assayed, these regions did not generate optimal results. Thus, genomic regions indicated as “qMSP poor performance” in our study specifically refer to CpG sites for which we encountered technical limitations after performing qMSP with alternative primer combinations and PCR conditions.

To address the reviewer’s comment and provide an alternative approach, we performed locus-specific bisulfite sequencing for all the genomic regions with poor qMSP performance (n=10; bisulfite sequencing primers sequences for each region can be found in **Supplementary Tables 3, 6, and 10**). Using genomic DNA and synthetic methylated and unmethylated universal controls (UMC and UUC) we were able to discriminate between methylated and unmethylated states of each of these loci (**examples provided below**). In our revised manuscript, we have included the new bisulfite sequencing protocol in the Methods section (**Page 27**), and indicated this alternative approach in the Results sections (**Page 10, Page 14, Page 19 and Supplementary Figures 3c, 7b, and 9b**).

This plot involves a summary of all the qMSP poor-performing regions assessed by the targeted bisulfite sequencing in our study (n=10 regions). These results are presented in the supplementary figures 3c, 7b,

and 9b of the revised manuscript. As a reference, Cytosines from CpG dinucleotides are highlighted in red in the wild type sequence, and replaced by an X in the bisulfite converted sequence, in which all the potential Cytosine to Thymine transitions are highlighted in green.

Compared to qMSP, the advantage of this approach is that the bisulfite-specific outer primers and the internal sequencing primers can be designed to recognize sequences flanking the high CpG density sequences. However, while qMSP-based DNA methylation data can be analyzed as a continuous variable, the locus-specific bisulfite sequencing data is often limited to a dichotomization of ‘methylated’ or ‘unmethylated’ states. Below we have evaluated a synthetic DNA methylation calibration curve (0%, 25%, 50%, 75%, and 100% methylated alleles per sample). As can be observed in this example, the extrapolation of bisulfite sequencing data to DNA methylation level is challenging, even after mathematical transformation of the ratios between the areas under the curves of methylated alleles (Cytosine residues) and unmethylated alleles (Thymine residues). Thus, due to this issue, we only recommend replacing the qMSP approach with bisulfite sequencing for highly informative, type-specific hypermethylated genomic regions where the dichotomized data would generate a scoring system with predictive potential, as we showed for the HM450K DNA methylation data (**Supplementary Fig. 2b-c**).

This plot presents the targeted bisulfite sequencing analysis of three genomic regions (one per BrainMETH classifier) using five samples with controlled DNA methylation levels (0%, 25%, 50%, 75%, and 100%) previously employed to generate the qMSP logarithmic equations. These results clearly show that bisulfite sequencing efficiently identifies the methylated status of the targeted alleles, but fails in providing an estimation of the percentage of DNAm, even when considering the areas under the curves of the Thymine (T) and Cytosine (C) variations.

3. The description of performance using differing numbers of markers is shown in figures 3a and 5a. In Figure 3a, it seems that a panel of >20 markers is sufficient for accurate classification, and in Figure 5a, one could argue that as little as three markers are sufficient. How were these features selected? In Figure 5a, it seems that only one marker, cg11730559, outperformed all other markers in the analyses. Were larger panels built upon

this initial marker? How were the levels of “feature importance” quantitatively measured? Is this a composite of specificity and sensitivity?

Response to Reviewer: Thank you for indicating this unclear section of our manuscript. In our revised version we have included substantial modifications to better explain the approaches used to generate, interpret, and test the DNA methylation classifiers. The reviewer is right that for the tissue of origin classifiers (**BrainMETH class B** in the manuscript; **Fig. 3a**), the high classification performance only starts to decline when using fewer than 20 genomic regions (decreasing about 30%, from 1.0 to 0.7 cross-validation performance). Conversely, in the BCBM-subtype classifier (**BrainMETH class C** in the manuscript; **Fig. 5a**) there is a moderate-to-good performance even when combining up to 10,000 genomic regions, which modestly decreases when using fewer than 10 markers (decreasing less than 10%, from 0.67 to 0.6 cross-validation performance). These observations can be explained by two facts: **1)** the sample size for the tissue of origin classifier is much larger than for the breast cancer subtype classifier, enabling more complex and predictive patterns to be learned from the data (n=94 vs n=28, respectively) and **2)** there is a much larger difference in global DNA methylation distribution, and thus discriminatory power, between cancers from different tissues than between different breast cancer subtypes. To assess this observation, in our revised manuscript, we compared the inner group distances (i.e. variance of a genomic region within one BM type or one BCBM subtype) with the distances between samples (i.e. variance of a genomic region between samples of different BM types or BCBM subtypes) and computed the F-ratios for each genomic region in the initial datasets. Supporting our observations, this analysis showed a significantly larger variance explained by the originating tissue of the brain metastasis than by the breast cancer molecular subtypes (mean F-Ratio: 10.02 ± 16.4 vs mean F-Ratio: 2.29 ± 3.35 , respectively; Wilcoxon test; P -value $< 2.2e-16$, **see boxplots below**). The effects of differences in global DNAm and sample size on classifier performance was clearly reflected in the final **BrainMETH** classifier C (BCBM molecular subtype), in which at least two genomic regions per BCBM molecular subtype have been combined to reach high sensitivity and specificity, instead of one genomic region per BM type, as required for the final **BrainMETH** classifier B (tissue of origin). To contextualize these dissimilar classification performances, we have indicated the variation in global DNA methylation distribution and sample size for each classifier in the Results section (**Page 17**) and reviewed the impact of these differences on the robustness of the final classifiers in the Discussion section (**Pages 22-23**).

The boxplots presented in this figure represent a measurement of the DNAm data variation according to the investigated categories. This analysis considered the variance of each genomic region in the initial datasets for groups containing brain metastasis types (BCBM, LCBM, and MBM) and BCBM subtypes (HR+/HER2-, HER2+, and HR-/HER2-). **The right plot** contains the data for the sum of the squares among the different samples in the dataset (SS_{between}) weighted to account for the differences in the sample sizes into the Mean Squares or variance explained by grouping (MS_{between}). **The middle plot** contains the data for the sum of the squares that represents the variation within the samples (SS_{within}) weighted to account for the differences in the sample sizes into the Mean Squares or variance unexplained by grouping (MS_{within}). **The left plot** contains the distribution of the resulting **F-Ratio** ($MS_{\text{between}}/MS_{\text{within}}$) which represents the variance explained by the grouping method in each case.

It is also an interesting observation that, in the BCBM subtype classifier, there is one marker (cg11730559) that outperforms all other markers, by at least doubling the feature importance of any of the top 5 most important genomic regions. However, this is in the context of a modest overall performance of the BCBM molecular subtypes. In fact, as it can be observed in the information provided in the revised **Figures 3a** and **5a**, the feature importance score of this genomic region is still significantly lower (Gini impurity score = 0.21; **Fig. 5c**) than the feature importance scores of markers involved in the tissue of origin classifier, which showed 13 genomic regions with a higher feature importance (Gini impurity score > 0.21; **Fig. 3c**). Several BCBM subtype classifiers included this marker (cg11730559). In fact, this marker is part of the top most differentially methylated genomic regions among all three BCBM molecular subtypes (**Supplementary Table 7**) and was selected as one of the 126 genomic regions whose HM450K-based DNA methylation levels were specifically associated with BCBM molecular subtypes (**Supplementary Table 9**), and useful in predicting the immunohistochemistry profile of BCBM specimens (**Fig. 4c-d**). Yet, due to the complexity of CpG site distribution, we were not able to design quality primer sets to survey its DNA methylation level by qMSP.

We apologize for the limited explanation of the feature selection algorithms in our original manuscript. Features were selected using a backward elimination procedure. We began with 10,000 predictive genomic regions, kept the most important half and repeated this procedure until

only one feature was left. The feature importance was measured as the Gini impurity score, the standard method used to calculate and report feature importance metrics in Random Forests algorithms⁵. The iterative backward elimination procedure was performed in a stratified cross-validation (CV) strategy. The feature importance scores reported in **Figures 3c** and **5c** are averaged across the (100) CV runs. In the revised version, we have added clarification of this approach in both the Results section (**Page 13** and **Page 18**) as well as the Methods section (**Page 30** and **Page 32**).

4. No specific genes were listed in the figures – only Illumina cg probe IDs. What are the gene regions identified from these analyses? Are these important for the biology of these tumors? What is the consequence of differential DNA methylation in terms of gene regulation? While gene expression aspects are not critical for identifying DNA methylation-based classifiers, an analysis of gene expression may help identify pathways for treatment or other diagnostic means.

Response to Reviewer: We agree with the reviewer's observation that DNA methylation marks have an important impact in the gene expression and provide novel, potentially actionable, therapeutic targets. In light of this, in our original manuscript, we included the genomic location, as well as the distance to the nearest gene for each of the identified genomic regions during our analyses (**Supplementary Tables 3, 6, and 10**). We apologize that this information was not clearly accessible, and we have specifically included statements to guide the readers to find the aforementioned information. Furthermore, we agree with the reviewer that while the identification of the most informative DNA methylation marks for the construction of classifiers is independent of nearby genes and gene expression, examining the impact epigenetic marks may have on these features could improve our understanding of these diseases. To address the inquiry regarding the potential consequences of differential DNA methylation in terms of gene regulation, we regret the fact that there is an insufficient collection of paired gene expression profiling and DNA methylation for metastatic brain tumors in both our cohort and other studies. However, the Genomic Regions Enrichment of Annotations Tool (GREAT) provides biological context by analyzing the annotations of proximal and distal genes⁶. In our original manuscript, we employed this approach to identify differentially affected pathways between primary and metastatic brain tumors, considering BM type specific hypomethylated genomic regions (**Page 9** and **Supplementary Fig. 1e**). In our revised manuscript, we have expanded the utilization of this tool to identify potentially differentially affected genes and pathway activation among the brain metastasis types (new **Supplementary Table 5** and **Page 11**). This analysis revealed enrichment of commonly affected pathways and gene networks, as well as changes that were BM type-specific. Based on these preliminary findings, specific studies would be required to investigate the relative importance of each of these regulatory gene networks and pathways for BM types and subtypes. Of note, due in part to the small numbers of differentially methylated genomic regions amongst the BCBM molecular subtypes (HR+/HER2- n=115, HER2+ n=70, and HR-/HER2- n=215), we found no significant enrichment for BCBM

molecular subtype specific gene pathways (threshold: hypergeometric test; FDR-corrected q-value =0.05). We have indicated this observation in the Results section (**Page 16**).

5. In Figure 2e, why were only a sub-cohort of women used in this analysis?

Response to Reviewer: Only a sub-cohort of BM specimens from female patients was included in this principal component analysis (PCA) in order to evaluate whether the gender of the patients with BCBM specimens (all females) was a factor in the striking separation of this cohort from the rest of the metastatic brain tumor specimens (patients with LCBM and patients with MBM).

Reviewer #3 (Remarks to the Author):

The current work by Javier Orozco et al. generated epigenetic profiling for the molecular classification of metastatic brain tumors, and particularly they constructed the DNAm-based brain classifier (BrainMETH) which has significant value in the molecular diagnosis of intracranial metastases.

While brain metastasis-defining epigenetic alterations which can be utilized to further develop DNA methylation profiling as a critical tool in the histomolecular stratification of patients with brain metastases, the work suffers from many limitations, per below.

Major concerns:

1. Figure 1: to identify intrinsic differences in the epigenetic profiles of metastatic brain tumors, the author generated DNAm signatures for 96 microdissected BM tissues. In the experiment, the author try to decrease potential biases associated with the gender of BM patients, they excluded 105,422 probes recognizing regions located in sex chromosomes or with proven cross-reactivity with sex chromosomes. This reviewer doesn't agree with conclusions. We think there are no need to exclude the probe located in sex chromosomes, because the majority breast cancer patients are female, and sometimes the prognosis/treatment response are very different among female and male lung cancer brain metastasis patients. Furthermore, ratios between male and female patients in every subtype of breast cancer or lung cancer are drastically different, therefore in these cases the gender influence could not be excluded. Lastly, Fig.1A simply describes which genomic regions were excluded from the analysis is irrelevant or difficult to unambiguously analyze. This is not a result, it is a technical issue and should be presented in the methods part only.

Response to Reviewer:

The reviewer is certainly right about the major differences in the gender distribution of metastatic brain tumors and its clinical-pathological implications. We agree that the exclusion of these genomic regions from the initial analysis restricts the ability to identify DNA methylation signatures with potential theranostic applications. However, our preliminary exploration of the DNA methylation data showed that the inclusion of sex-specific genomic regions certainly biased the classification performance. As indicated by the reviewer, since all the breast cancer brain metastasis specimens included in our study were resected from female patients, the classifier distinguished breast cancer samples from melanoma and lung cancer samples based simply on gender differences, instead of by the type of disease. From the genomic regions excluded due to their propensity to bind to or have cross reactivity with the sex chromosomes, we identified 4,229 which recognize the X-chromosome with a high degree of specificity. Examining these regions with dimensionality reduction analyses, we observed a substantial influence of the patient's gender on spatial

distribution of the BM specimens (**see plots below**). Thus, by excluding the genomic regions located in or having potential cross-reactivity with sex chromosomes, we tried to identify types of brain metastases based *not* on sex differences, but on tissue-specific DNA methylation signals. Following the reviewer’s rationale and due to the homogeneity on the gender distribution, the construction of the DNA methylation classifiers to predict breast cancer brain metastasis molecular subtypes was performed including genomic regions located on the X-chromosome. In fact, one of the genomic regions whose DNA methylation level exhibited high predictive potential to discriminate HER2+ breast cancer brain metastases by HM450K microarray and qMSP analysis is located at the promoter region of the *CFP* gene in the X-chromosome (HER2+C; chrX:47,483,258; **Supplementary Table 10**). This clarification has been included in our revised manuscript (**Page 15**).

We also agree with the reviewer that the filtering steps for problematic genomic regions provided in **Figure 1a** is not a result, but a part of the method employed for this study. However, due to the journal format, which includes the Results section before the Methods section, we believe it is more helpful to present this information before providing the results generated from this initial dataset.

This figure includes the resulting principal component analyses for all the brain metastasis (BM) specimens included in our study considering the DNA methylation level of genomic regions located in the X-chromosome with good quality for design and HM450K evaluation. The analysis was performed to compare the spatial distribution of BM specimens according to this DNA methylation data and highlighted according to the patient's gender (Upper panels: left panel shows PC1 vs PC2 and right panel shows PC1 and PC3) or BM type (BCBM, LCBM, and MBM; Lower panels: left panel shows PC1 vs PC2 and right panel shows PC1 and PC3). These plots indicate a substantial influence of the patient's gender on spatial distribution of the BM specimens.

2. Figure 3: authors identified the genomic regions difference among breast/lung/melanoma brain metastasis for the prediction of the tumor of origin. However, did authors verify genomic region differences between the brain metastasis with primary sites (breast cancer, lung cancer and melanoma cancer) using TCGA data, plus their own patient data?

Response to Reviewer: This is an excellent observation that added an interesting new application to our study. In this regard, we used the differentially methylated regions evaluated in our brain metastasis samples in a cohort of equivalent primary tumor samples from TCGA (n= 791) in two ways. **1)** We employed the top 100 most predictive regions identified in the brain metastasis tissue of origin classifier (**BrainMETH class B** in the manuscript) to generate a principal component analysis (PCA) and a hierarchical cluster analysis (HCL) for the TCGA primary tumor samples. These analyses showed that the DNA methylation levels of regions identified in our brain metastasis samples separate the primary tumors according to the tissue of origin (PCA: the first three components explained 75.5 percent of the cumulative variance and HCL: bootstrap resampling showed 100% of support for the separation between the cluster containing the primary melanomas and the cluster containing primary breast and lung carcinomas and 78% of support for the separation between the cluster containing most of the primary breast tumors and the cluster containing most of the primary lung tumor specimens; **Supplementary Fig. 6a-b**). **2)** In our revised manuscript, we additionally built a Random Forest classifier of tissue of origin employing the corresponding TCGA primary tumor specimen cohort, and compared the overall behavior of both classifiers. Importantly, supporting our observations in the multidimensionality reduction analyses, we found a large and statistically significant overlap in predictive genomic regions shared by both classifiers (hypergeometric test; P -value $<2.8e-23$). Together, these data suggest that genomic region differences between BMs are comparable to genomic region differences between their corresponding primary tumors. This new information has been included in the revised manuscript (**Page 13**).

The reviewer additionally points out another interesting potential analysis: comparing specific primary tumor types to their brain metastasis counterparts (i.e. primary lung cancer to lung cancer BM), may reveal distinguishing DNA methylation patterns. Such epigenetic differences between primary tumors and their respective metastases could shed light on tumor evolution,

allowing inferences to be drawn regarding the influence of DNA methylation on metastatic progression, potentially opening new avenues of research on novel therapeutic approaches. These processes are no doubt important to fully understanding disease etiology. However, in our current study, the main focus was to investigate potential diagnostic applications, rather than evolutionary or theranostic DNA methylation markers, which we believe to be a more immediate need to aid the pathological classification. In addition, in order to more accurately address the reviewer's suggestion of investigating DNA methylation markers of progression, we believe future studies should include a cohort of patients with paired primary and brain metastasis specimens, a comparison that was not possible to perform during our current study. The novel brain metastasis clinical-demographic (new **Supplementary Table 1**) and DNA methylation data (GEO series records: GSE44661 and GSE108576, reviewer access token: upubgkywjxytjiv) provided in our revised study would significantly contribute to the development of such relevant comparisons.

Second (Fig.3A), classifiers to predict tumor of origin for BM were identified among DNA methylation patterns. Using data set training and testing, this yielded fantastic results, both in terms of sensitivity and specificity. It also showed that using combination of 40 or more (up to 10,000) classifiers a maximal (approximately 98%) prediction performance is achieved. In light of these results, Fig.5A is hard to understand. Here, classifiers to predict BCBM molecular subtypes are derived and tested with the same approach as in Fig.3A. Results showed that, on average, the predictive power of one classifier alone is the same as the combination of 3, 5, 10, up to 10,000. This is very confusing and an explanation for this phenomenon should have been provided in the text. Furthermore, the predictive power is much less striking, range 30% to 80%. Is this still valuable clinically?

Response to Reviewer: Thank you for pointing out these important differences between the performances of the tissue of origin (**BrainMETH** class B in the manuscript) classifier (**Fig. 3a**) and the BCBM-subtype (**BrainMETH** class C in the manuscript) classifier (**Fig. 5a**) and the need to discuss them in more detail. As the reviewer correctly observed, the classifiers generated to predict the tissue of origin of the different type of brain metastases had a substantially better performance than the classifiers generated to identify the breast cancer molecular subtypes. As we mention in pages 9 and 10 of this letter, the differences in classification performance can be explained by two facts: **1)** the sample size for the tissue of origin classifier is much larger than for the BCBM-subtype classifier, enabling more complex and predictive patterns to be learned from the data (n=94 vs n=28, respectively), and **2)** there is a significantly larger difference in global DNA methylation distribution, and thus discriminatory power, between BM specimens from different tumor types than between BCBM specimens from different subtypes. In our revised manuscript, we have computed these global differences by comparing the inner group distances (i.e. variance of a genomic region within one BM type or one BCBM subtype) with the distances between samples (i.e. variance of a genomic region between samples of different BM types or BCBM subtypes) and reported the differential distribution of the F-ratios for each genomic region in the initial datasets (The boxplots

presented on page 10 of this letter summarize our findings in this regard). This new information if provided in the Results section (**Page 17**). Thus, in view of the modest overall classification performance of this BCBM subtype classifiers, specifically for the underrepresented (HR-/HER2-) BCBM subtype (**Fig. 5b**), we agree with the reviewer's concern about the immediate clinical value of this classifier. As a consequence of this modest classification performance, we have strengthened this classifier by assessing at least two genomic regions per BCBM subtype by qMSP, which allowed us to reach good sensitivity and specificity (AUCs >0.9) in the identification of the three BCBM subtypes (**Fig. 5f**). Yet, we agree that the robustness of this classifier must be further improved using additional cases before being applied in the clinic. However, we believe data presented here serve as a proof of concept that BCBM subtypes can be distinguished based on DNAm profiles. We have added clarification of this issue in the revised Discussion section (**Pages 22-23**).

What is the difference between Fig.5B and Fig.5F?

Response to Reviewer: Thank you for pointing this unclear section of our study. Figure 5b represents the averaged sensitivity and specificity for the classifiers generated using the Illumina HM450K microarray DNA methylation data for each of the breast cancer brain metastasis molecular subtype. Figure 5f, on the other hand, represents the receiver operating curves and area under the curves generated from the combinations of the qMSP analysis of the genomic regions selected and validated for each breast cancer molecular subtype. In our revised manuscript, we have clarified the differences between the two plots (**Page 23**).

Third, (Fig.3D), genomic regions which predicted well BM tissue origin yielded very different results when applied to primary tumors of the same tissues, e.g., breast cancer left and right panels, lung right panel). It is conceivable that BM of breast cancer have unique features compared to primary breast cancer. However, this result and the potential impact on patient-treatment decisions were not discussed.

Response to Reviewer: This is an excellent observation. After carefully reviewing these analyses, we noted that in our original submission we inadvertently switched the labels of the BCBM and LCBM groups in the bottom boxplots (with TCGA data) in Figure 3d and also in Figure 5d. We have therefore carefully re-run the analyses and revised the labeling of all the figures in the manuscript and confirmed this was the only error in the figures. Therefore, we found a substantial agreement between the DNA methylation patterns for brain metastases and primary tumors, which is now clearly represented by these boxplots (**Fig. 3d** and **Fig. 5d**). We agree with the reviewer that BM type and subtype specific genomic regions with differential methylation between primary and brain metastases may be informative for disease management and understanding of cancer biology. However, this would warrant a different study design mostly focused on the comparison of paired primary and metastatic brain tumors. In our revised manuscript, we have discussed the need for

these additional studies to address these interesting aspects and applications of DNA methylomes (Page 24).

3. Figure 4: authors used the DNAm profiling to predict BCBM molecular subtypes. However, did authors analyze the DNAm profiling to predict lung cancer metastasis/melanoma cancer brain metastasis subtype?

Response to Reviewer: This is an interesting point. Unlike breast cancer molecular subtypes, which are defined and usually diagnosed by specific gene and protein expression signatures, melanoma and lung cancer therapeutic or molecular subtypes are defined by gene mutations (i.e. activating mutations affecting the *BRAF*, *NRAS*, *EGFR*, *ROS1*, *KRAS* or *ALK* genes, amongst others) and, in some cases, gene fusions (i.e. *ALK-EML4*, *FIG-ROS1*, or *CD74-ROS1* genes, amongst others) that impact therapy selection. As a matter of practicality, we believe that the identification of cutaneous melanoma and lung cancer brain metastasis therapeutic subtypes should be performed by targeted fusion screenings⁷ or targeted gene sequencing, as we have previously shown⁸. Therefore, while new DNA methylation classifiers for lung and melanoma brain metastasis could add relevant clinical-pathological information, we prioritized the identification of breast cancer brain metastases therapeutic subtypes due to associated prevalent diagnostic challenges. However, in our revised manuscript, we provide a clinical-demographic and pathological annotation for each patient, allowing researchers to further investigate DNA methylation signatures and classifiers for any of the included tumor types which may be of interest (new **Supplementary Table 1** and **Supplementary Data 1**).

4. Figures 5 & 6: authors found the breast cancer brain metastasis molecular subtype-specific DNA methylation profiles and used them to aid in the identification of breast cancer brain metastasis molecular subtype. Are these DNA methylation signatures helpful to distinguish different type of primary breast cancer type? Or differences of DNAm profiles among different subtypes of primary breast cancer types are similar to differences in DNA profiles among different three BCBM subtypes? Further (Fig.5D), same concerns are raised as in Fig.3D: genomic regions which worked to predict BM molecular subtype yielded drastically distinct results when applied to primary tumors. Accordingly, is using these regions for molecular subtype determinations still valid and applicable?

Response to Reviewer: We apologize for the confusion caused by the incorrect ordering of the boxplots of the primary tumors presented in Figure 5d, as mentioned before, we corrected this in the revised version (**Fig. 5d**). Yet, as mentioned earlier, due to the modest overall classification performance, we strengthened this classifier by assessing at least two genomic regions per BCBM molecular subtype with qMSP, which allowed us to reach AUC higher than 0.9 for each subtype (**Fig. 5f**, **Supplementary Table 10**, indicated in **Page 19** and discussed in **Page 23**). As the reviewer also suggests, we tested the performance of the BCBM subtype classifier in a large cohort

of TCGA primary breast cancer specimens (n=643) by applying dimensionality reduction approaches using the principal component analysis (PCA) for the top 100, 50, 30, 15, 10, and 5 most informative BCBM genomic regions (**Supplementary Fig. 8a**). Unlike our observations for the tissue of origin classifier, DNA methylation profiles discriminating between BCBM specimens exhibited a minimal ability to separate primary breast cancer tumors according to their respective molecular subtypes. Specifically, PCA using these DNA methylation signatures only explained a mean of 58.50 ± 3.2 percent of the cumulative variance (**Page 18**). Therefore, our data suggest that genomic regions informative distinguishing breast cancer molecular subtypes differ between primary tumors and brain metastases. As such, the practical solution for circumventing these differences is to identify the most informative set of genomic regions using the primary breast cancer specimens DNA methylomes. We have discussed these observations in our revised manuscript (**Page 23**).

5. Did authors ever compared the DNAm profiling between GBM and three different brain metastasis cancer type separately?

Response to Reviewer: This is an interesting suggestion. To address this question and provide more insights about the underlying epigenetic mechanisms of these diseases, we compared the DNA methylomes of each brain metastasis type with GBM specimens. By doing so, we observed that the genomic regions whose DNA methylation level significantly separated GBM from all the BMs are also useful to separate GBM from each of the specific BM type. This new information is included in the revised manuscript (**Page 8** and **Supplementary Fig. 1a**).

Minor concerns:

1. The bar graphs for feature importance in Fig.3C and Fig.5C do not have scale numbers.

Response to Reviewer: Thank you for pointing the lack of scale on these plots; both figures were corrected in the revised version of the manuscript (**Fig. 3c** and **Fig. 5c**).

2. Authors are recommended to discuss further the use of results achieved, e.g. the refinement of BM classification, correlations to aggressiveness, stage of disease, etc.

Response to Reviewer: The reviewer points out important additional clinical features of interest, which may potentially be assessed by examining DNA methylation patterns of the metastatic brain tumors included in our study. However, these features are outside the scope of our current study, which was aimed to investigate diagnostic rather than prognostic markers for brain metastasis. As we discussed above for the study of the impact of prior treatment approaches on the brain metastasis DNA methylomes, the investigation of these interesting prognostic aspects will require a

specific framework to be examined and reported properly. Thus, a separate study with a population meeting the statistical power requirements, is needed, ideally within a prospective clinical trial, in order to generate conclusive data about these important clinical end-points. In our revised manuscript, we have included all the clinical and demographic data for the entire patient cohort in our study, including follow-up intervals, outcomes, and therapeutic interventions, hoping that this additional information will complement ongoing studies addressing these critical issues (new **Supplementary Table 1**).

References

- 1 Capper, D. *et al.* DNA methylation-based classification of central nervous system tumours. *Nature* **555**, 469-474 (2018).
- 2 Newton, Y. *et al.* TumorMap: Exploring the Molecular Similarities of Cancer Samples in an Interactive Portal. *Cancer Res.* **77**, e1111-e1114 (2017).
- 3 Varadhachary, G. R. & Raber, M. N. Cancer of unknown primary site. *N. Engl. J. Med.* **371**, 757-765 (2014).
- 4 Ceccarelli, M. *et al.* Molecular Profiling Reveals Biologically Discrete Subsets and Pathways of Progression in Diffuse Glioma. *Cell* **164**, 550-563 (2016).
- 5 Ishwaran, H. The Effect of Splitting on Random Forests. *Mach. Learn.* **99**, 75-118 (2015).
- 6 McLean, C. Y. *et al.* GREAT improves functional interpretation of cis-regulatory regions. *Nat. Biotechnol.* **28**, 495-501 (2010).
- 7 Leighl, N. B. *et al.* Molecular testing for selection of patients with lung cancer for epidermal growth factor receptor and anaplastic lymphoma kinase tyrosine kinase inhibitors: American Society of Clinical Oncology endorsement of the College of American Pathologists/International Association for the study of lung cancer/association for molecular pathology guideline. *J. Clin. Oncol.* **32**, 3673-3679 (2014).
- 8 Marzese, D. M. *et al.* DNA methylation and gene deletion analysis of brain metastases in melanoma patients identifies mutually exclusive molecular alterations. *Neuro Oncol.* **16**, 1499-1509 (2014).

Reviewer #1 (Remarks to the Author):

The authors have carried out significant new analyses and revisions, and in my opinion, have adequately addressed my prior concerns.

Reviewer #2 (Remarks to the Author):

In my opinion, the authors have addressed my concerns, so I choose to accept the revised manuscript for publication.

REVIEWERS' COMMENTS NCOMMS-18-03458B:

Reviewer #1 (Remarks to the Author):

The authors have carried out significant new analyses and revisions, and in my opinion, have adequately addressed my prior concerns.

Reviewer #2 (Remarks to the Author):

In my opinion, the authors have addressed my concerns, so I choose to accept the revised manuscript for publication.

Response to Reviewers: We thank the reviewers for their constructive criticisms and valuable suggestions to our original version of the study. We believe that after adding the required analyzes and re-contextualization of the results, the manuscript has significantly improved. We are glad that the reviewers found our new analyzes, data, and experiments adequate.